# Robust Learning Rate Selection for Stochastic Optimization via Splitting Diagnostic

**Matteo Sordello**  *matteo.sordello91@gmail.com*
*Department of Statistics and Data Science*
*Wharton School, University of Pennsylvania*
*Philadelphia, PA, USA*

**Niccolò Dalmasso**  *niccolo.dalmasso@gmail.com*
*Department of Statistics & Data Science*
*Carnegie Mellon University*
*Pittsburgh, PA, USA*

**Hangfeng He**  *hangfeng@seas.upenn.edu*
*Department of Computer and Information Science*
*University of Pennsylvania*
*Philadelphia, PA, USA*

**Weijie Su**  *suw@wharton.upenn.edu*
*Department of Statistics and Data Science*
*Wharton School, University of Pennsylvania*
*Philadelphia, PA, USA*

**Reviewed on OpenReview:** *https://openreview.net/forum?id=3PbxuMNQkp*

## Abstract

This paper proposes SplitSGD, a new dynamic learning rate schedule for stochastic optimization. This method decreases the learning rate for better adaptation to the local geometry of the objective function whenever a *stationary* phase is detected, that is, the iterates are likely to bounce at around a vicinity of a local minimum. The detection is performed by splitting the single thread into two and using the inner product of the gradients from the two threads as a measure of stationarity. Owing to this simple yet provably valid stationarity detection, SplitSGD is easy-to-implement and essentially does not incur additional computational cost than standard SGD. Through a series of extensive experiments, we show that this method is appropriate for both convex problems and training (non-convex) neural networks, with performance compared favorably to other stochastic optimization methods. Importantly, this method is observed to be very robust with a set of default parameters for a wide range of problems and, moreover, can yield better generalization performance than other adaptive gradient methods such as Adam.

## 1 Introduction

Many machine learning problems boil down to finding a minimizer $\theta^* \in \mathbb{R}^d$ of a risk function taking the form

$$F(\theta) = \mathbb{E}\left[f(\theta, Z)\right], \tag{1}$$

where $f$ denotes a loss function, $\theta$ is the model parameter, and the *random* data point $Z = (X, y)$ contains a feature vector $X$ and its label $y$. In the case of a finite population, for example, this problem is reduced to the empirical minimization problem. The touchstone method for minimizing (1) is stochastic gradient descent (SGD). Starting from an initial point $\theta_0$, SGD updates the iterates according to

$$\theta_{t+1} = \theta_t - \eta_t \cdot g(\theta_t, Z_{t+1}) \tag{2}$$

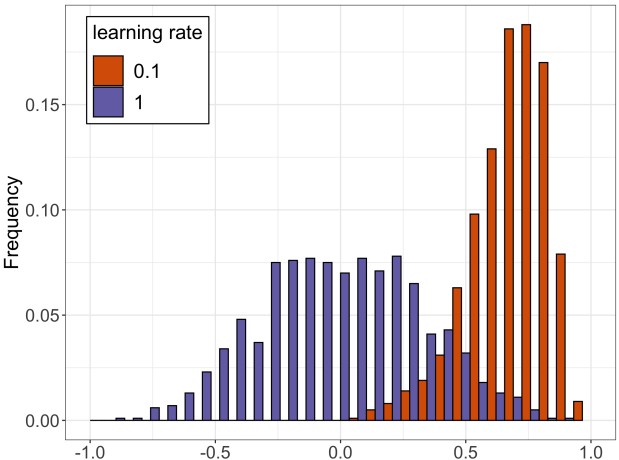

Figure 1: Normalized dot product of averaged noisy gradients over 100 iterations. Stationarity depends on the learning rate: $\eta = 1$ corresponds to stationarity (purple), while $\eta = 0.1$ corresponds to non stationarity (orange). Details in Section 2.

for $t \geq 0$, where $\eta_t$ is the learning rate, $\{Z_t\}_{t=1}^{\infty}$ are i.i.d. copies of $Z$ and $g(\theta, Z)$ is the (sub-) gradient of $f(\theta, Z)$ with respect to $\theta$. The noisy gradient $g(\theta, Z)$ is an unbiased estimate for the true gradient $\nabla F(\theta)$ in the sense that $\mathbb{E}[g(\theta, Z)] = \nabla F(\theta)$ for any $\theta$.

The convergence rate of SGD crucially depends on the *learning rate*—often recognized as "the single most important hyper-parameter" in training deep neural networks (Bengio, 2012)—and, accordingly, there is a vast literature on how to *decrease* this fundamental tuning parameter for improved convergence performance. In the pioneering work of Robbins & Monro (1951), the learning rate $\eta_t$ is set to $O(1/t)$ for convex objectives. Later, it was recognized that a slowly decreasing learning rate in conjunction with iterate averaging leads to a faster rate of convergence for strongly convex and smooth objectives (Ruppert, 1988; Polyak & Juditsky, 1992). More recently, extensive effort has been devoted to incorporating preconditioning/Hessians into learning rate selection rules (Duchi et al., 2011; Dauphin et al., 2015; Tan et al., 2016). Among numerous proposals, a simple yet widely employed approach is to repeatedly halve the learning rate after performing a *pre-determined* number of iterations (see, for example, Bottou et al., 2018).

In this paper, we introduce a new variant of SGD that we term *SplitSGD* with a novel learning rate selection rule. At a high level, our new method is motivated by the following fact: an optimal learning rate should be adaptive to the *informativeness* of the noisy gradient $g(\theta_t, Z_{t+1})$. Roughly speaking, the informativeness is higher if the true gradient $\nabla F(\theta_t)$ is relatively large compared with the noise $\nabla F(\theta_t) - g(\theta_t, Z_{t+1})$ and vice versa. On the one hand, if the learning rate is too small with respect to the informativeness of the noisy gradient, SGD makes rather slow progress. On the other hand, the iterates would bounce around a region of an optimum of the objective if the learning rate is too large with respect to the informativeness. The latter case corresponds to a stationary phase in stochastic optimization (Murata, 1998; Chee & Toulis, 2018), which necessitates the *reduction* of the learning rate for better convergence. Specifically, let $\pi_\eta$ be the stationary distribution for $\theta$ when the learning rate is constant and set to $\eta$. From (2) one has that $\mathbb{E}_{\theta \sim \pi_\eta}[g(\theta, Z)] = 0$, and consequently that

$$\mathbb{E}[\langle g(\theta^{(1)}, Z^{(1)}), g(\theta^{(2)}, Z^{(2)}) \rangle] = 0 \tag{3}$$

for $\theta^{(1)}, \theta^{(2)} \overset{i.i.d.}{\sim} \pi_\eta$ and $Z^{(1)}, Z^{(2)} \overset{i.i.d.}{\sim} Z$.

SplitSGD differs from other stochastic optimization procedures in its *robust* stationarity phase detection, which we refer to as the *Splitting Diagnostic*. In short, this diagnostic runs two SGD threads initialized at the same iterate using *independent* data points (refers to $Z_{t+1}$ in (2)), and then performs hypothesis testing to determine whether the learning rate leads to a stationary phase or not. The effectiveness of the Splitting Diagnostic is illustrated in Figure 1, which reveals different patterns of dependence between the two SGD threads with difference learning rates. Loosely speaking, in the stationary phase (in purple), the two SGD

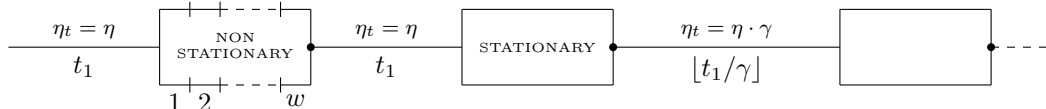

Figure 2: The architecture of SplitSGD. The initial learning rate is $\eta$ and the length of the first single thread is $t_1$. If the diagnostic does not detect stationarity, the length and learning rate of the next thread remain unchanged. If stationarity is observed, we decrease the learning rate by a factor $\gamma$ and proportionally increase the length.

threads behave as if they are independent due to a large learning rate, and SplitSGD subsequently decreases the learning rate by some factor. In contrast, strong positive dependence is exhibited in the non stationary phase (in orange) and, thus, the learning rate remains the same after the diagnostic. In essence, the robustness of the Splitting Diagnostic is attributed to its adaptivity to the *local geometry* of the objective, thereby making SplitSGD a *tuning-insensitive* method for stochastic optimization. Its strength is confirmed by our experimental results in both convex and non-convex settings. In the latter, SplitSGD showed robustness with respect to the choice of the initial learning rate, and remarkable success in improving the test accuracy and avoiding overfitting compared to classic optimization procedures.

## 1.1 Related work

There is a long history of detecting stationarity or non-stationarity in stochastic optimization to improve convergence rates (Yin, 1989; Pflug, 1990; Delyon & Juditsky, 1993; Murata, 1998). Perhaps the most relevant work in this vein to the present paper is Chee & Toulis (2018), which builds on top of Pflug (1990) for general convex functions. Specifically, this work uses the running sum of the inner products of successive stochastic gradients for stationarity detection. However, this approach does not take into account the strong correlation between consecutive gradients and, moreover, is not sensitive to the local curvature of the current iterates due to unwanted influence from prior gradients. In contrast, the splitting strategy, which is akin to HiGrad (Su & Zhu, 2018), allows our SplitSGD to concentrate on the current gradients and leverage the regained independence of gradients to test stationarity. Lately, Yaida (2019) and Lang et al. (2019) derive a stationarity detection rule that is based on gradients of a mini-batch to tune the learning rate in SGD with momentum while Pesme et al. (2020) base their diagnostic on the distance between the current iterate and the last iterate where learning rate reduction happened.

From a different angle, another related line of work is concerned with the relationship between the informativeness of gradients and the mini-batch size (Keskar et al., 2016; Yin et al., 2017; Li et al., 2017; Smith et al., 2017). Among others, it has been recognized that the optimal mini-batch size should be adaptive to the local geometry of the objective function and the noise level of the gradients, delivering a growing line of work that leverage the mini-batch gradient variance for learning rate selection (Byrd et al., 2012; Balles et al., 2016; Balles & Hennig, 2017; De et al., 2017; Zhang & Mitliagkas, 2017; McCandlish et al., 2018).

## 2 The SplitSGD algorithm

In this section, we first develop the Splitting Diagnostic for stationarity detection, reported in Algorithm 1, followed by the introduction of SplitSGD, detailed in Algorithm 2.

### 2.1 Diagnostic via Splitting

Intuitively, the stationarity phase occurs when two independent threads with the same starting point are *no longer* moving along the same direction. This intuition is the motivation for our Splitting Diagnostic, which is presented in Algorithm 2 and described in what follows. We call $\theta_0$ the initial value, even though later it will often have a different subscript based on the number of iterations already computed before starting the diagnostic. From the starting point, we run two SGD threads, each consisting of $w$ windows of length $l$. For

---

**Algorithm 1 Diagnostic**$(\eta, w, l, q, \theta^{in})$

---

1:  $\theta_0^{(1)} = \theta_0^{(2)} = \theta^{in}$

2:  **for** $i = 1, ..., w$ **do**

3:      **for** $k = 1, 2$ **do**

4:          **for** $j = 0, ..., l - 1$ **do**

5:              $\theta_{(i-1)\cdot l+j+1}^{(k)} = \theta_{(i-1)\cdot l+j}^{(k)} - \eta \cdot g_{(i-1)\cdot l+j}^{(k)}$

6:          **end for**

7:          $\bar{g}_i^{(k)} = (\theta_{(i-1)\cdot l+1}^{(k)} - \theta_{i\cdot l}^{(k)})/l \cdot \eta.$

8:      **end for**

9:      $Q_i = \langle \bar{g}_i^{(1)}, \bar{g}_i^{(2)} \rangle$

10: **end for**

11: **if** $\sum_{i=1}^{w}(1 - \mathrm{sign}\,(Q_i))/2 \geq q \cdot w$ **then**

12:     **return** $\left\{ \theta_D = (\theta_{w\cdot l}^{(1)} + \theta_{w\cdot l}^{(2)})/2, \; T_D = S \right\}$

13: **else**

14:     **return** $\left\{ \theta_D = (\theta_{w\cdot l}^{(1)} + \theta_{w\cdot l}^{(2)})/2, \; T_D = N \right\}$

15: **end if**

---

each thread $k = 1, 2$, we define $g_t^{(k)} = g(\theta_t^{(k)}, Z_{t+1}^{(k)})$ and the iterates are

$$\theta_{t+1}^{(k)} = \theta_t^{(k)} - \eta \cdot g_t^{(k)}, \tag{4}$$

where $t \in \{0, ..., wl - 1\}$. A similar splitting strategy can be also found in the HiGrad procedure introduced in Su & Zhu (2018). There, the authors use the estimates produced by several parallel threads to obtain a confidence interval around $\theta^*$, keeping the learning rate constant and using a decorrelating procedure. Here, instead, on every thread we compute the average noisy gradient in each window, indexed by $i = 1, ..., w$, which is

$$\bar{g}_i^{(k)} := \frac{1}{l} \sum_{j=1}^{l} g_{(i-1)\cdot l+j}^{(k)} = \frac{\theta_{(i-1)\cdot l+1}^{(k)} - \theta_{i\cdot l+1}^{(k)}}{l \cdot \eta}. \tag{5}$$

The length $l$ of each window has the same function as the mini-batch parameter in mini-batch SGD (Li et al., 2014), in the sense that a larger value of $l$ aims to capture more of the true signal by averaging out the errors. At the end of the diagnostic, we have stored two vectors, each containing the average noisy gradients in the windows in each thread.

**Definition 1** *For $i = 1, ..., w$, we define the gradient coherence with respect to the starting point of the Splitting Diagnostic $\theta_0$, the learning rate $\eta$, and the length of each window $l$, as*

$$Q_i(\theta_0, \eta, l) = \langle \bar{g}_i^{(1)}, \bar{g}_i^{(2)} \rangle. \tag{6}$$

*We will drop the dependence from the parameters and refer to it simply as $Q_i$.*

The gradient coherence expresses the relative position of the average noisy gradients, and its sign indicates whether the SGD updates have reached stationarity. In fact, if in the two threads the noisy gradients are pointing on average in the same direction, it means that the signal is stronger than the noise, and the dynamic is still in its transient phase. On the contrary, as (3) suggests, when the gradient coherence is on average very close to zero, and it also assumes negative values thanks to its stochasticity, this indicates that the noise component in the gradient is now dominant, and stationarity has been reached. Of course these values, no matter how large $l$ is, are subject to some randomness. Our diagnostic then considers the signs of $Q_1, ..., Q_w$ and returns a result based on the proportion of negative $Q_i$. One output is a boolean value $T_D$, defined as

---

**Algorithm 2 SplitSGD**$(\eta, w, l, q, B, t_1, \theta_0, \gamma)$

---

1: $\eta_1 = \eta$

2: $\theta_1^{in} = \theta_0$

3: **for** $b = 1, ..., B$ **do**

4:    Run SGD with constant step size $\eta_b$ for $t_b$ steps, starting from $\theta_b^{in}$

5:    Let the last update be $\theta_b^{last}$

6:    $D_b = \textbf{Diagnostic}(\eta_b, w, l, q, \theta_b^{last})$

7:    $\theta_{b+1}^{in} = \theta_{D_b}$

8:    **if** $T_{D_b} = S$ **then**

9:        $\eta_{b+1} = \gamma \cdot \eta_b$ and $t_{b+1} = \lfloor t_b/\gamma \rfloor$

10:   **else**

11:       $\eta_{b+1} = \eta_b$ and $t_{b+1} = t_b$

12:   **end if**

13: **end for**

---

follows:

$$T_D = \begin{cases} S & \text{if} \quad \sum_{i=1}^{w}(1 - \text{sign}\,(Q_i))/2 \geq q \cdot w \\ N & \text{if} \quad \sum_{i=1}^{w}(1 - \text{sign}\,(Q_i))/2 < q \cdot w. \end{cases} \tag{7}$$

where $T_D = S$ indicates that stationarity has been detected, and $T_D = N$ means non-stationarity. The parameter $q \in [0, 1]$ controls the tightness of this guarantee, being the smallest proportion of negative $Q_i$ required to declare stationarity. In addition to $T_D$, we also return the average last iterate of the two threads as a starting point for following iterations. We call it $\theta_D := (\theta_{w \cdot l}^{(1)} + \theta_{w \cdot l}^{(2)})/2$. The gradient coherence has a similar flavor to the `pflug` diagnostic that is used in Chee & Toulis (2018). There, a running sum of the dot products of consecutive gradients from a single thread is stored, and stationarity is declared when such statistic becomes negative. A downside of such strategy is that the initial positive dot products can have a large impact on the rest of the procedure. Because of an ill selected initial learning rate, it can happen that the running sum of the dot products approaches zero very slowly, making the stationarity detection less practical. We show in Section 4.1 how the Splitting Diagnostic improves on the `pflug` diagnostic by avoiding such problem.

## 2.2 The Algorithm

The Splitting Diagnostic can be employed in a more sophisticated SGD procedure, which we call SplitSGD. We start by running the standard SGD with constant learning rate $\eta$ for a fixed number of iterations $t_1$. Then, starting from $\theta_{t_1}$, we use the the next $2lw$ updates to verify if stationarity has been reached, using the Splitting Diagnostic. If stationarity is not detected, the next single thread has the same length $t_1$ and learning rate $\eta$ as the previous one. On the contrary, if $T_D = S$, we realize that it is the time to decrease the learning rate by a factor $\gamma \in (0, 1)$. At the same time we also increase the length of the single thread by a factor $1/\gamma$, as suggested by Bottou et al. (2018) in their procedure, later analyzed under the name of SGD$^{1/2}$ by Chee & Toulis (2018). Notice that, if $q = 0$, then the learning rate gets deterministically decreased after each diagnostic since every Splitting Diagnostic will return $T_D = S$. On the other extreme, if we set $q = 1$, then the procedure maintains constant learning rate with high probability, since the only chance to decay the learning rate happens when all the gradient coherences are negative. Figure 2 illustrates what happens when the first diagnostic does not detect stationarity, but the second one does. SplitSGD puts together two crucial aspects: it employs the Splitting Diagnostic at deterministic times, but it does not deterministically decreases the learning rate. We will see in Section 4 how both of these features combine to give advantage over existing methods. It is important to notice that SplitSGD does not add relevant computational cost compared to classic SGD. Although there are two threads that are used in some parts of the SplitSGD procedure, we i) decide their length in advance so that they do not use the majority of the iterates compared to the single

threads and ii) use averaging at the end of the Splitting Diagnostic, which makes the second thread useful in the computation of the minimizer, as we will see in the experiment section. Moreover, the computation of the gradient coherence is small compared to the cost of training the model. A detailed explanation of SplitSGD is presented in Algorithm 2.

## 3   Theoretical Guarantees for Stationarity Detection

This section develops theoretical guarantees for the validity of our learning rate selection. Specifically, in the case of a relatively small learning rate, we can imagine that, if the number of iterations is fixed, the SGD updates are not too far from the starting point, so the stationary phase has not been reached yet. On the other hand, however, when $t \to \infty$ and the learning rate is fixed, we would like the diagnostic to tell us that we have reached stationarity, since we know that in this case the updates will oscillate around $\theta^*$. Our first assumption concerns the convexity of the function $F(\theta)$. It will not be used in Theorem 2, in which we focus our attention on a neighborhood of $\theta_0$.

**Assumption 3.1** *The function $F$ is strongly convex, with convexity constant $\mu > 0$. For all $\theta_1, \theta_2$,*

$$F(\theta_1) \geq F(\theta_2) + \langle \nabla F(\theta_2), \theta_1 - \theta_2 \rangle + \frac{\mu}{2} \|\theta_1 - \theta_2\|^2$$

*and also $\|\nabla F(\theta_1) - \nabla F(\theta_2)\| \geq \mu \cdot \|\theta_1 - \theta_2\|$.*

**Assumption 3.2** *The function $F$ is smooth, with smoothness parameter $L > 0$. For all $\theta_1, \theta_2$,*

$$\|\nabla F(\theta_1) - \nabla F(\theta_2)\| \leq L \cdot \|\theta_1 - \theta_2\|.$$

We said before that the noisy gradient is an unbiased estimate of the true gradient. The next assumption that we make is on the distribution of the errors.

**Assumption 3.3** *We define the error in the evaluation of the gradient in $\theta_{t-1}$ as*

$$\epsilon_t := \epsilon(\theta_{t-1}, Z_t) = g(\theta_{t-1}, Z_t) - \nabla F(\theta_{t-1}) \tag{8}$$

*and the filtration $\mathcal{F}_t = \sigma(Z_1, ..., Z_t)$. Then $\epsilon_t \in \mathcal{F}_t$ and $\{\epsilon_t\}_{t=1}^{\infty}$ is a martingale difference sequence with respect to $\{\mathcal{F}_t\}_{t=1}^{\infty}$, which means that $\mathbb{E}[\epsilon_t|\mathcal{F}_{t-1}] = 0$. The covariance of the errors satisfies*

$$\sigma_{\min} \cdot I \preceq \mathbb{E}\left[\epsilon_t \epsilon_t^T \mid \mathcal{F}_{t-1}\right] \preceq \sigma_{\max} \cdot I, \tag{9}$$

*where $0 < \sigma_{\min} \leq \sigma_{\max} < \infty$ for any $\theta$.*

Our last assumption is on the noisy functions $f(\theta, Z)$ and on an upper bound on the moments of their gradient. We do not specify $m$ here since different values are used in the next two theorems, but the range for this parameter is $m \in \{2, 4\}$.

**Assumption 3.4** *Each function $f(\theta, Z)$ is convex, and there exists a constant $G$ such that $\mathbb{E}\left[\|g(\theta_t, Z_{t+1})\|^m \mid \mathcal{F}_t\right] \leq G^m$ for any $\theta_t$.*

Note that Assumption 3.3 imposes that the eigenvalue of a rank-one matrix is finite, which is a relatively mild assumption, and Assumption 3.4 correspond to assumption H7 in Moulines & Bach (2011). We first show that there exists a learning rate sufficiently small such that the standard deviation $\text{sd}(Q_i)$ of any gradient coherence is arbitrarily small compared to its expectation, and the expectation is positive when $\theta_{t_1+l}$ is not very far from $\theta_0$. This implies that the probability of any gradient coherence to be negative, $\mathbb{P}(Q_i < 0)$, is extremely small, which means that the Splitting Diagnostic will return $T_D = N$ with high probability.

**Theorem 2** *If Assumptions 3.2, 3.3 and 3.4 with $m = 4$ hold, $\|\nabla F(\theta_0)\| > 0$ and we run $t_1$ iterations before a Splitting Diagnostic with $w$ windows of length $l$, then for any $i \in \{1, ..., w\}$ we can set $\eta$ small enough to guarantee that*

$$\text{sd}(Q_i) \leq C_1(\eta, l) \cdot \mathbb{E}[Q_i],$$

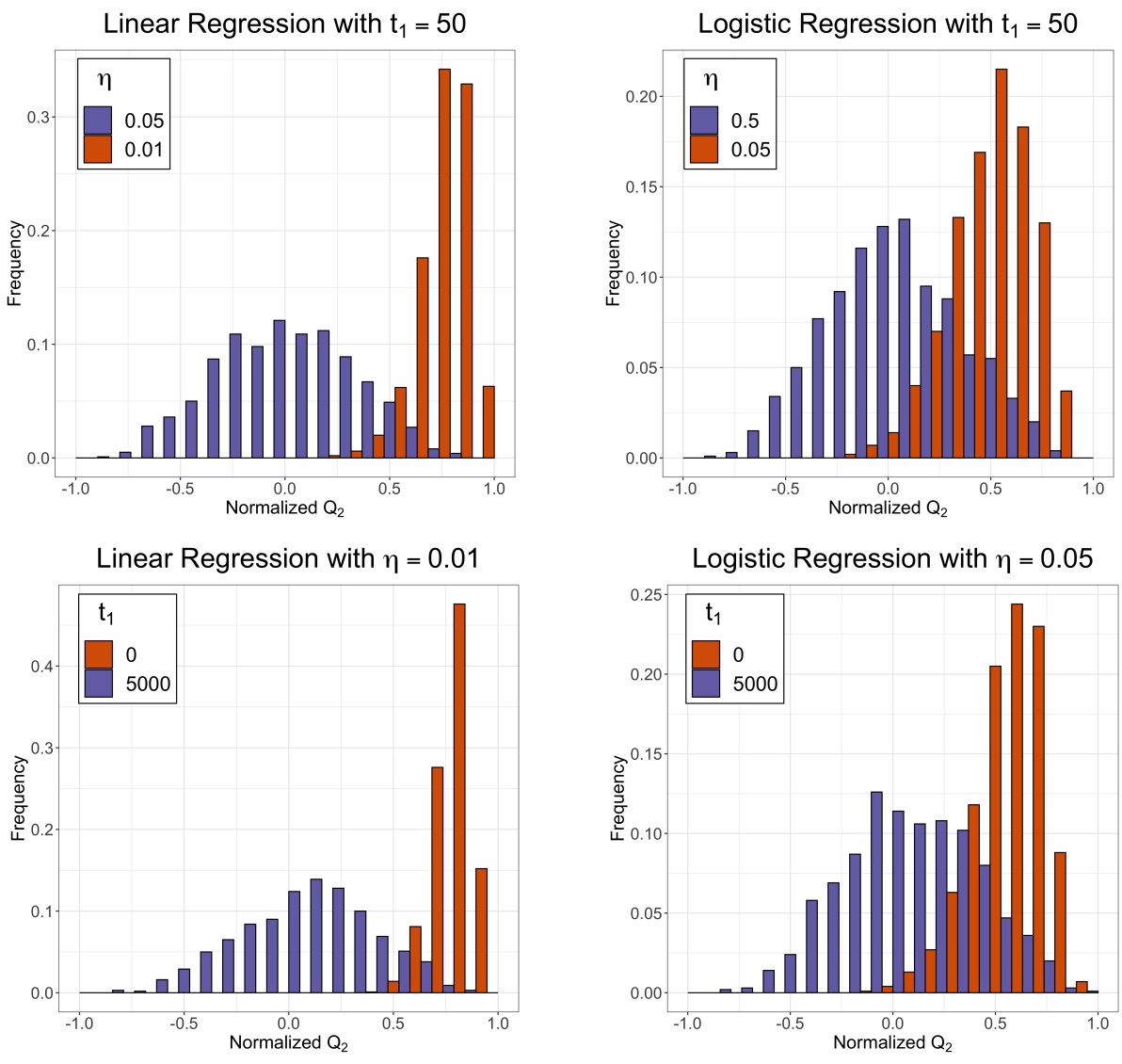

Figure 3: Histogram of the gradient coherence $Q_i$ (for the second pair of windows, normalized) of the Splitting Diagnostic for linear and logistic regression. The two top panels show the behavior in Theorem 2, the two bottom panels the one in Theorem 3. In orange we see non stationarity, while in purple a distribution that will return stationarity for an appropriate choice of $w$ and $q$.

where $C_1(\eta, l) = O(1/\sqrt{l}) + O(\sqrt{\eta(t_1 + l)})^1$. *The proof is in Appendix F.*

In the two top panels of Figure 3 we provide a visual interpretation of this result. When the starting point of the SGD thread is sufficiently far from the minimizer $\theta^*$ and $\eta$ is sufficiently small, then all the mass of the distribution of $Q_i$ is concentrated on positive values, meaning that the Splitting Diagnostic will not detect stationarity with high probability. In particular we can use Chebyshev inequality to get a bound for $\mathbb{P}(Q_i < 0)$ of the following form:

$$\mathbb{P}(Q_i < 0) \leq \mathbb{P}(|Q_i - \mathbb{E}[Q_i]| > \mathbb{E}[Q_i])$$
$$\leq \mathrm{sd}(Q_i)^2/\mathbb{E}[Q_i]^2 \leq C_1(\eta, l)^2$$

---

[1]$C_1(\eta, l)$ also depends on $\|\nabla F(\theta_0)\|, d, \sigma_{\max}, L$ and $G$

Note that to prove Theorem 2 we do not need to use the strong convexity Assumption 3.1 since, when $\eta(t_1 + l)$ is small, $\theta_{t_1+l}$ is not very far from $\theta_0$. In the next Theorem we show that, if we let the SGD thread before the diagnostic run for long enough and the learning rate is not too big, then the splitting diagnostic output is $T_D = S$ with probability that can be made arbitrarily high. This is consistent with the fact that, as $t_1 \to \infty$, the iterates will start oscillating in a neighborhood of $\theta^*$.

**Theorem 3** *If Assumptions 3.1, 3.2, 3.3 and 3.4 with $m = 2$ hold, then for any $\eta \leq \frac{\mu}{L^2}$, $l \in \mathbb{N}$ and $i \in \{1, ..., w\}$, as $t_1 \to \infty$ we have*

$$|\mathbb{E}[Q_i]| \leq C_2(\eta) \cdot \mathrm{sd}(Q_i),$$

*where $C_2(\eta) = C_2 \cdot \eta + o(\eta)$. The proof is in Appendix G.*

The result of this theorem is confirmed by what we see in the bottom panels of Figure 3. There, most of the mass of $Q_i$ is on positive values if $t_1 = 0$, since the learning rate is sufficiently small and the starting point is not too close to the minimizer. But when we let the first thread run for longer, we see that the distribution of $Q_i$ is now centered around zero, with an expectation that is much smaller than its standard deviation. An appropriate choice of $w$ and $q$ makes the probability that $T_D = S$ arbitrarily big. In the proof of Theorem 3, we make use of a result that is contained in Moulines & Bach (2011) and then subsequently improved in Needell et al. (2014), representing the dynamic of SGD with constant learning rate. For completeness we report its proof in Appendix E.

**Lemma 4** *If Assumptions 3.1, 3.2, 3.3 and 3.4 with $m = 2$ hold, and $\eta \leq \frac{\mu}{L^2}$, then for any $t \geq 0$*

$$\mathbb{E}\left[\|\theta_t - \theta^*\|^2\right] \leq \left(1 - 2\eta(\mu - L^2\eta)\right)^t \cdot \mathbb{E}\left[\|\theta_0 - \theta^*\|^2\right]$$
$$+ \frac{G^2\eta}{\mu - L^2\eta}.$$

The simulations in Figure 3 show us that, once stationarity is reached, the distribution of the gradient coherence is fairly symmetric and centered around zero, so its sign will be approximately a coin flip. In this situation, if $l$ is large enough, the count of negative gradient coherences is approximately distributed as a Binomial with $w$ number of trials, and 0.5 probability of success. Then we can set $q$ to control the probability of making a type I error – rejecting stationarity after it has been reached – by making $\frac{1}{2^w} \sum_{i=0}^{q \cdot w - 1} \binom{w}{i}$ sufficiently small. Notice that a very small value for $q$ makes the type I error rate decrease but makes it easier to think that stationarity has been reached too early. In the Appendix A we provide a simple visual interpretation to understand why this trade-off gets weaker as $w$ becomes larger, while we show in Figure 4 with a simple experiment how the Splitting Diagnostic is detecting stationarity with the correct probability once stationarity is actually been reached. What we see is that when $w$ is small the agreement is not perfect, but as it gets larger the theoretical and observed probabilities nearly overlap perfectly.

Finally, we provide a result on the convergence of SplitSGD. We leave for future work to prove the convergence rate of SplitSGD, which appears to be a very challenging problem.

**Proposition 5** *If Assumptions 3.1, 3.2, 3.3 and 3.4 with $m = 2$ hold, and $\eta \leq \frac{\mu}{L^2}$, then SplitSGD is guaranteed to converge with probability tending to 1 as the number of diagnostics $B \to \infty$.*

**Proof** We first notice that the averaging at the end of each diagnostic can be ignored, and replaced by simply considering each diagnostic as a single thread made of $wl$ iterates. For the first diagnostic, for example, we have that

$$\mathbb{E}\left[\|\theta_{D_1} - \theta^*\|^2\right] \leq \mathbb{E}\left[\left\|\frac{\theta_{t_1+wl}^{(1)} + \theta_{t_1+wl}^{(2)} - 2\theta^*}{2}\right\|^2\right]$$
$$= \frac{1}{4}\mathbb{E}\left[\|\theta_{t_1+wl}^{(1)} - \theta^*\|^2\right] + \frac{1}{4}\mathbb{E}\left[\|\theta_{t_1+wl}^{(2)} - \theta^*\|^2\right]$$
$$+ \frac{1}{2}\mathbb{E}\left[\langle\theta_{t_1+wl}^{(1)} - \theta^*, \theta_{t_1+wl}^{(2)} - \theta^*\rangle\right]$$
$$\leq \mathbb{E}\left[\|\theta_{t_1+wl}^{(1)} - \theta^*\|^2\right]$$

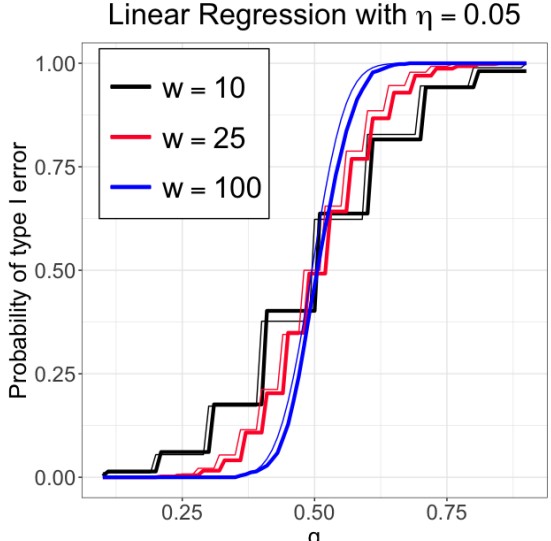
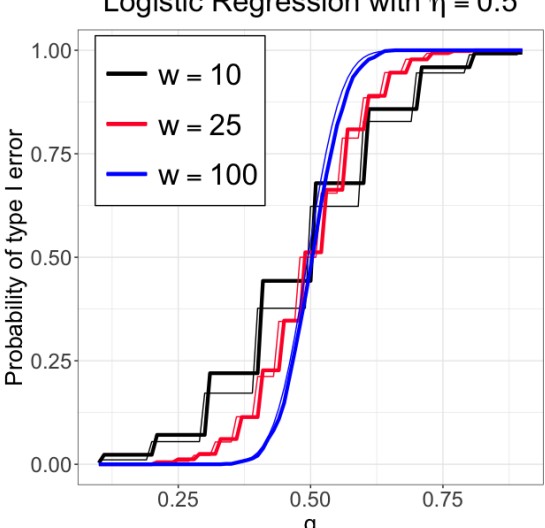

Figure 4: The probability of making a type I error using the Splitting Diagnostic (thick lines) closely matches with the respective theoretical probability (thin lines), in both linear and logistic regression settings. In both settings we considered 1000 experiments for each value of $w$ and we initialize $\theta_0$ to be close to $\theta^*$, with $l = 10$ and $\eta$ sufficiently large to guarantee stationarity.

where we have used the fact that each thread is identically distributed, together with the Cauchy-Schwarz inequality. The same inequality, with appropriate indexes, is true for all the diagnostics.

Our proof is now divided in two parts. First we show that, in the extreme case where each diagnostic detects stationarity deterministically, the learning rate does not decay too fast and we still have convergence to $\theta^*$. Then we prove that eventually the learning rate decreases to zero when the number of diagnostics goes to infinity. We initially notice that

$$\left(1 - 2\eta\gamma^b(\mu - L^2\eta\gamma^b)\right)^{t_1/\gamma^b} \leq e^{-2\eta(\mu - L^2\eta)t_1} =: c_1$$

where $c_1 \in (0, 1)$. We also have

$$\frac{G^2\eta\gamma^b}{\mu - L^2\eta\gamma^b} \leq \frac{G^2\eta\gamma^b}{\mu - L^2\eta} =: c_2 \cdot \gamma^b$$

We define $L_b$ to be the expected square distance from the minimizer, $\mathbb{E}\left[\|\theta_{D_b} - \theta^*\|^2\right]$, at the end of the $b^{th}$ diagnostic, and $L_0 = \mathbb{E}\left[\|\theta_0 - \theta^*\|^2\right]$. If the learning rate decreases deterministically, then we have that after the $b^{th}$ diagnostic, the learning rate is $\eta\gamma^b$ and the length of the single thread is $\lfloor t_1/\gamma^b \rfloor$. By recursion, using Lemma 4 in the main text, we have that

$$
\begin{aligned}
L_{b+1} &\leq \left(1 - 2\eta\gamma^b(\mu - L^2\eta\gamma^b)\right)^{t_1/\gamma^b} \cdot L_b + \frac{G^2\eta\gamma^b}{\mu - L^2\eta\gamma^b} \\
&\leq c_1 \cdot L_b + c_2 \cdot \gamma^b \\
&\leq c_1^{b+1} \cdot L_0 + c_2 \cdot \sum_{i=0}^{b} \gamma^{b-i} c_1^i \\
&\leq c_1^{b+1} \cdot L_0 + c_2 \cdot b \cdot \max\{\gamma, c_1\}^b
\end{aligned}
$$

Since $\gamma, c_1 \in (0, 1)$, this proves that $L_b \to 0$ as the number of diagnostics $b \to \infty$.

To prove that it is impossible for the learning rate to remain fixed on a certain value for infinite many iterations, we show that the probability that the learning rate reaches a point where it never decreases is zero.

We assume by contradiction that there exists a point in the SplitSGD procedure where the learning rate is $\eta^*$ and, from that moment on, it is never reduced again. Following Dieuleveut et al. (2017), we know that the Markov chain $\{\theta_t\}$ in (2) with constant learning rate $\eta^*$ will converge in distribution to its stationary distribution $\pi_{\eta^*}$. This means that

$$\sup_{s,t\geq T} \|\mathbb{E}[\theta_t] - \mathbb{E}[\theta_s]\| \to 0 \qquad \text{as } T \to \infty \tag{10}$$

and if we let $s = t + 1$ we realise that $\|\mathbb{E}[g(\theta_t, Z_{t+1})]\| \to 0$ as $t \to \infty$. Notice that also the Markov chain $\{g(\theta_t, Z_{t+1})\}$ converges to a stationarity distribution when $\{\theta_t\}$ does, so we can use the Central Limit Theorem for Markov chains (Maxwell & Woodroofe, 2000) to get that

$$\frac{1}{\sqrt{l}} \sum_{j=1}^{l} g(\theta_{t+j}, Z_{t+j+1}) \xrightarrow{d} N(0, \sigma^2) \qquad \text{as } l \to \infty \tag{11}$$

where $\sigma^2 > 0$. We are now going to use the fact that $\text{sign}(Q_i) = \text{sign}(l \cdot Q_i)$. Thanks to (11) we can now write

$$\begin{aligned}
l \cdot Q_i &= \left\langle \frac{1}{\sqrt{l}} \sum_{j=1}^{l} g^{(1)}_{t+(i-1)l+j}, \frac{1}{\sqrt{l}} \sum_{j=1}^{l} g^{(2)}_{t+(i-1)l+k} \right\rangle \\
&= \langle X_1 + o_p(1), X_2 + o_p(1) \rangle \\
&= \langle X_1, X_2 \rangle + o_p(1)
\end{aligned}$$

where $X_1, X_2$ are independent $N(0, \sigma^2)$ (the independence being true for $l \to \infty$ and $i = 2, ..., w$) and the $o_p(1)$ are defined as $l \to \infty$. Since $l \cdot Q_i$ is approximately distributed as $\langle X_1, X_2 \rangle$, which has mean zero and positive variance, then for any choice of $q < (w-1)/w$ we know that there is a positive probability $\alpha > 0$ that the proportion of negative gradient coherences observed is greater than $q$, which means that stationarity is detected. The probability that the learning rate $\eta^*$ never decays is then bounded above by $\lim_{b \to \infty} (1 - \alpha)^b = 0$, so the learning rate gets eventually reduced with probability 1. ∎

## 4 Experiments

We now compare the Splitting Diagnostic and SplitSGD procedure with other diagnostic and optimization techniques in both convex and non-convex settings.

### 4.1 Convex Objective

The setting is described in details in Appendix A. We use a feature matrix $X \in \mathbb{R}^{n \times d}$ with standard normal entries and $n = 1000$, $d = 20$ and $\theta^*_j = 5 \cdot e^{-j/2}$ for $j = 1, ..., 20$. The key parameters are $t_1 = 4, w = 20, l = 50$ and $q = 0.4$. For both linear and logistic regression, we use the streaming version of SGD where the batch size is set to 1, meaning that each data point in the training set is observed individually. A sensitivity analysis is in Section 4.3.

**Comparison between splitting and pflug diagnostic.** In the top panels of Figure 5 we compare the Splitting Diagnostic with the pflug Diagnostic introduced in Chee & Toulis (2018). The boxplots are obtained running both diagnostic procedures from a starting point $\theta_0 = \theta_s + \epsilon'$, where $\epsilon' \sim N(0, 0.01 I_d)$ is multivariate Gaussian and $\theta_s$ has the same entries of $\theta^*$ but in reversed order, so $\theta_{s,j} = 5 \cdot e^{-(d-j)/2}$ for $j = 1, ..., 20$. Each experiment is repeated 100 times. For the Splitting Diagnostic, we run SplitSGD and declare that stationarity has been detected at the first time that a diagnostic gives result $T_D = S$, and output the number of epochs up to that time. For the pflug diagnostic, we stop when the running sum of dot products used in the procedure becomes negative at the end of an epoch. The maximum number of epochs is 1000, and the red horizontal bands represent the approximate values for when we can assume that stationarity has been reached, based on when the loss function of SGD with constant learning rate stops decreasing. We can see that the result of the Splitting Diagnostic is close to the truth, while the pflug Diagnostic incurs the

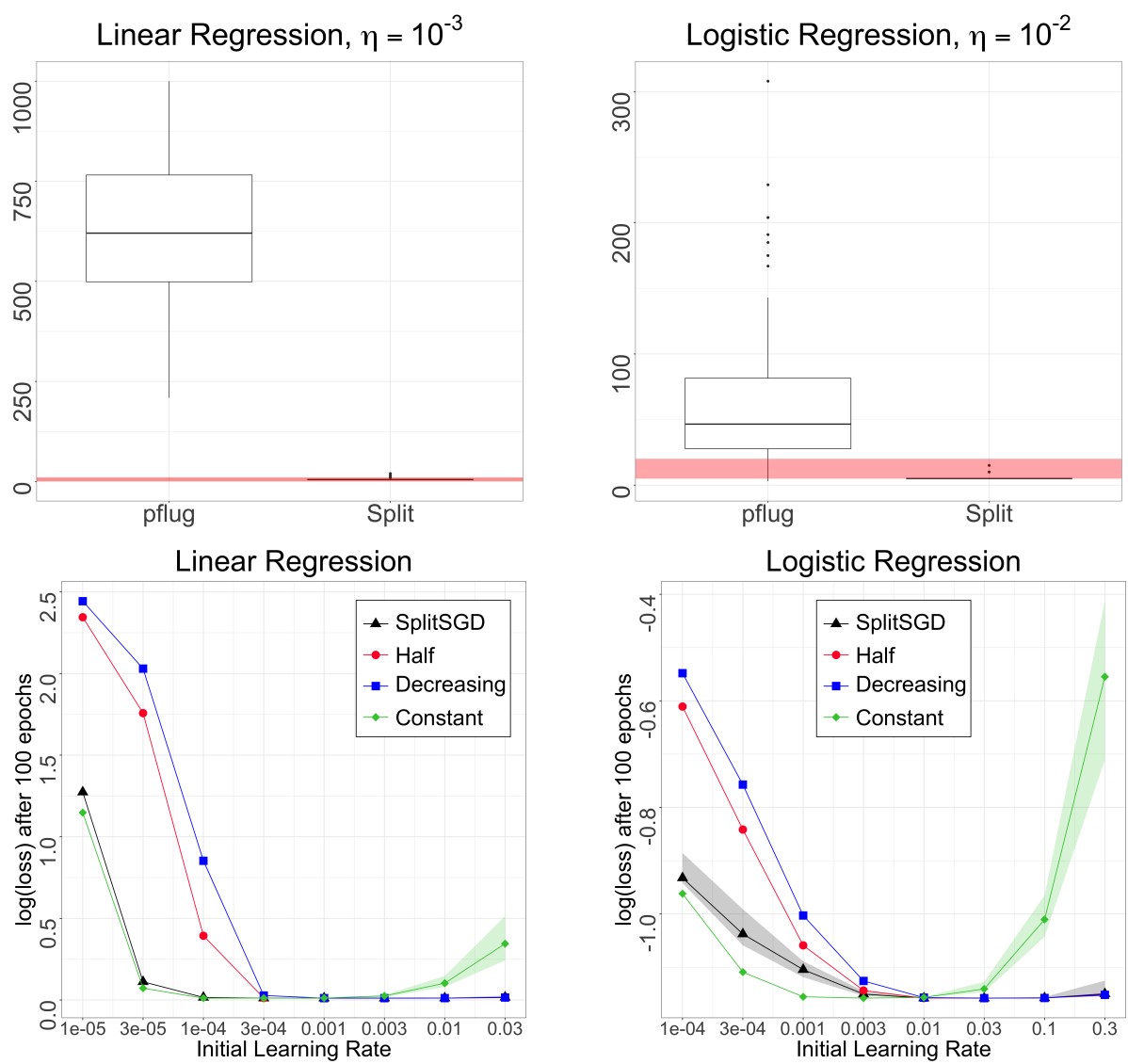

Figure 5: (Top) Comparison between Splitting and `pflug` Diagnostics on linear and logistic regression. The y-axis represents the epochs and the red bands are the epochs where stationarity should be detected, while the boxplots represent the distribution of when the method actually detects stationarity. The `pflug` Diagnostics incurs in the risk of waiting too long after stationarity is reached, while the Splitting Diagnostic does not as a checkpoint is set every fixed number of iterations. (Bottom) comparison of the log(loss) achieved after 100 epochs between SplitSGD, SGD$^{1/2}$ (Half) and SGD with constant or decreasing learning rate on linear and logistic regression. More details are in Section 4.1.

risk of waiting for too long, when the initial dot products of consecutive noisy gradients are positive and large compared to the negative increments after stationarity is reached. The Splitting Diagnostic does not have this problem, as a checkpoint is set every fixed number of iterations. The previous computations are then discarded, and only the new learning rate and starting point are stored. In Appendix B we show more configurations of learning rates and starting points.

**Comparison between SplitSGD and other optimization procedures.** Here we set the decay rate to the standard value $\gamma = 0.5$, and compare SplitSGD with SGD with constant learning rate $\eta$, SGD with decreasing learning rate $\eta_t \propto 1/\sqrt{t}$ (where the initial learning rate is set to $20\eta$), and SGD$^{1/2}$ (Bottou et al., 2018), where the learning rate is halved deterministically and the length of the next thread is double that of

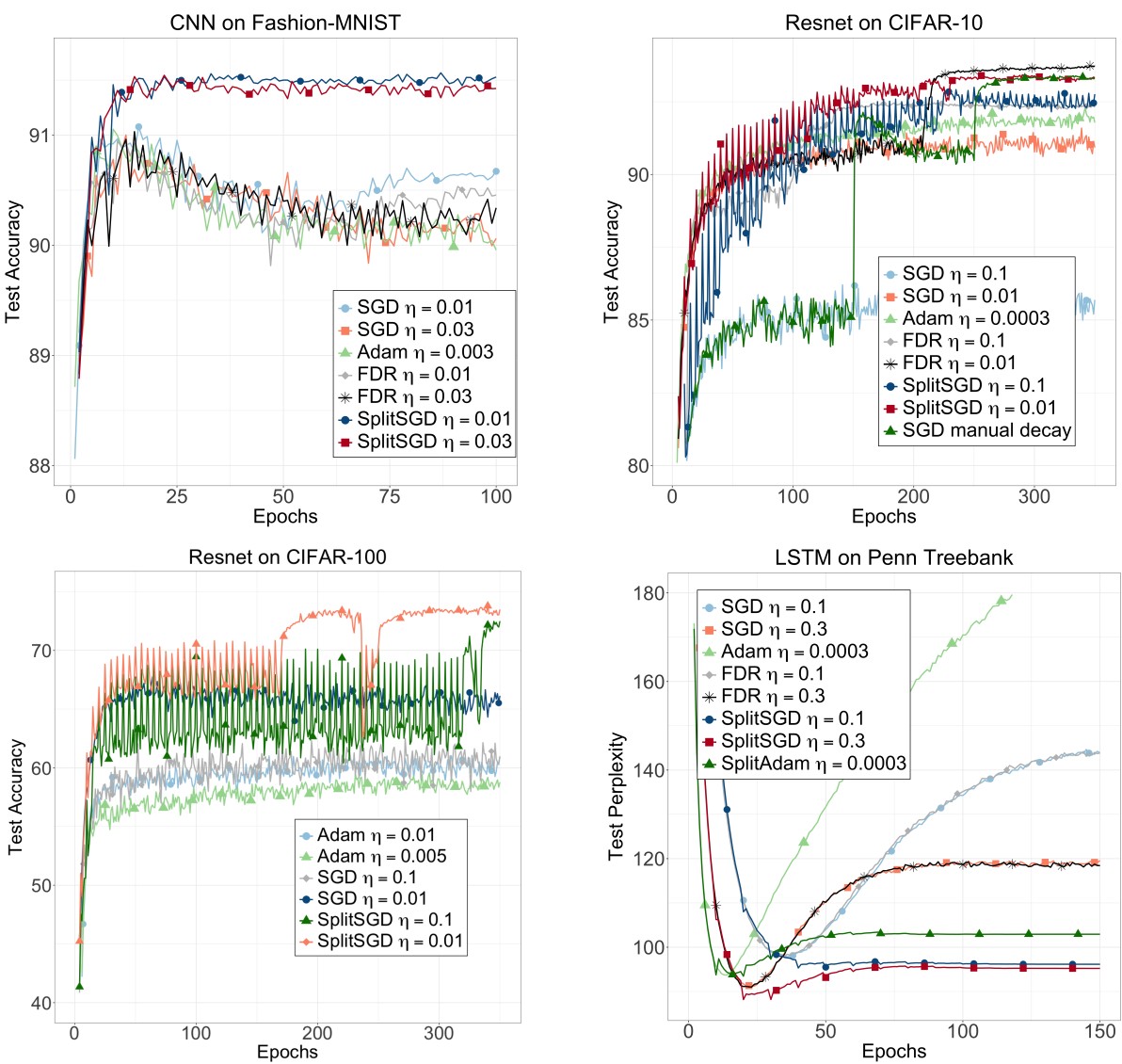

Figure 6: Performance of SGD, Adam, FDR and SplitSGD in training various neural networks across different tasks and datasets: (top left) a convolutional neural network on a 10-class computer vision task on Fashion-MNIST, (top right) a Resnet-18 on a 10-class computer vision task on CIFAR-10, (bottom left) a Resnet-18 on a 100-class computer vision task on CIFAR-100, and (bottom right) an LSTM on a NLP prediction task on Penn Treebank. SplitSGD proved to be beneficial in (i) better robustness to the choice of initial learning rates, (ii) achieving higher test accuracy when possible, and (iii) reducing the effect of overfitting. Details of each plot are in Section 4.2.

the previous one. For SGD$^{1/2}$ we set the length of the initial thread to be $t_1$, the same as for SplitSGD. In the bottom panels of Figure 5 we report the log of the loss that we achieve after 100 epochs for different choices of the initial learning rate. It is clear that keeping the learning rate constant is optimal when its initial value is small, but becomes problematic for large initial values. On the contrary, deterministic decay can work well for larger initial learning rates but performs poorly when the initial value is small. Here, SplitSGD shows its robustness with respect to the initial choice of the learning rate, performing well on a wide range of initial learning rates.

### 4.2 Deep Neural Networks

To train deep neural networks, instead of using the simple SGD with a constant learning rate inside the SplitSGD procedure, we adopt SGD with momentum (Qian, 1999), where the momentum parameter is set to 0.9. SGD with momentum is a popular choice in training deep neural networks (Sutskever et al., 2013), and when the learning rate is constant, it still exhibits both transient and stationary phase. We introduce three more differences with respect to the convex setting: (i) the gradient coherences are defined for each layer of the network separately, then counted together to globally decay the learning rate for the whole network, (ii) the length of the single thread is not increased if stationarity is detected, and (iii) we consider the default parameters $q = 0.25$ and $w = 4$ for each layer. We expand on these differences in Appendix C. As before, the length of the Diagnostic is set to be one epoch, and $t_1 = 4$. We compare SplitSGD against SGD with momentum and two other optimization methods widely used in practice. Adam (Kingma & Ba, 2014), which has been developed as an improvement over AdaGrad (Duchi et al., 2011) and RMSprop (Tieleman et al., 2012), stores the decaying averages of the first and second moment of the past gradients and uses them to compute an adaptive learning rate for each parameter. FDR (Yaida, 2019), instead, uses a stationarity detection rule based on two fluctuation-dissipation relations to decay the constant learning rate of SGD with momentum. We train our models using a range of different learning rates for all these methods, and report the ones that show the best results. Notice that, although $\eta = 3e-4$ is the popular default value for Adam, this method is still sensitive to the choice of the learning rate, so the best performance can be achieved with other values. For FDR, we tested each setting with the parameter t_adaptive $\in \{100, 1000\}$, which gave similar results. It has also been proved that SGD generalizes better than Adam (Keskar & Socher, 2017; Luo et al., 2019). We show that in many situations SplitSGD, using the same default parameters, can outperform both. In Figure 6 we report the average results of 5 runs. In Figure 12 in the appendix we consider the same plot but also add 90% confidence bands, omitted here for better readability.

**Convolutional neural networks (CNNs).** We consider a CNN with two convolutional layers and a final linear layer trained on the Fashion-MNIST dataset (Xiao et al., 2017). We set $\eta \in \{1e-2, 3e-2, 1e-1\}$ for SGD and SplitSGD, $\eta \in \{1e-2, 1e-1\}$ for FDR and $\eta \in \{3e-4, 1e-3, 3e-3, 1e-2\}$ for Adam. The batch size is 64 across all models. In the first panel of Figure 6 we see the interesting fact that SGD, FDR and Adam all show clear signs of overfitting, after reaching their peak in the first 20 epochs. SplitSGD, on the contrary, does not incur in this problem, but for a combined effect of the averaging and learning rate decay is able to reach a better overall performance without overfitting. We also notice that SplitSGD is very robust with respect to the choice of the initial learning rate, and that its peak performance is better than the one of any of the competitors.

**Residual neural networks (ResNets) on Cifar-10.** We consider a 18-layer ResNet[2] and evaluate it on the CIFAR-10 dataset (Krizhevsky et al., 2009). We use the initial learning rates $\eta \in \{1e-3, 1e-2, 1e-1\}$ for SGD and SplitSGD, $\eta \in \{1e-2, 1e-1\}$ for FDR and $\eta \in \{3e-5, 3e-4, 3e-3\}$ for Adam, and also consider the SGD procedure with manual decay that consists in setting $\eta = 1e-1$ and then decreasing it by a factor 10 at epoch 150 and 250. For consistency, we set the batch size to 128 across all models. In the second panel of Figure 6 we see a classic behavior for SplitSGD. The averaging after the diagnostics makes the test accuracy peak, but the improvement is only momentary as the learning rate is not decreased. When the decay happens, the peak is maintained and the fluctuations get smaller. We can see that SplitSGD, with both initial learning rate $\eta = 1e-2$ and $\eta = 1e-1$ is better than both SGD and Adam and that one setting achieves the same final test accuracy of the manually tuned method in less epochs. The FDR method is showing excellent performance when $\eta = 0.01$ and a worse result when $\eta = 0.1$. In Appendix D we see a similar plot obtained with the neural network VGG19.

**Residual neural networks (ResNets) on Cifar-100.** To show the performance of SplitSGD on a more complex classification task, we have also evaluated 18-layer ResNet on the CIFAR-100 dataset. Here we only compared SplitSGD with SGD and Adam, and for all considered $\eta \in \{5e-3, 1e-2, 3e-1, 1e-1\}$. As with CIFAR-10, we set the batch size to 128 across all models. In the third panel of 6 we reported the two best runs for each method, and from those we can see that SplitSGD is able to achieve a higher peak accuracy compared to the other methods. Here the Splitting Diagnostic appears to work slightly better when $\eta = 0.01$,

---

[2]More details at `https://pytorch.org/docs/stable/torchvision/models.html`.

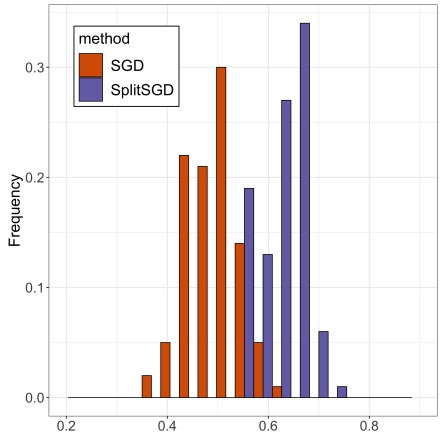 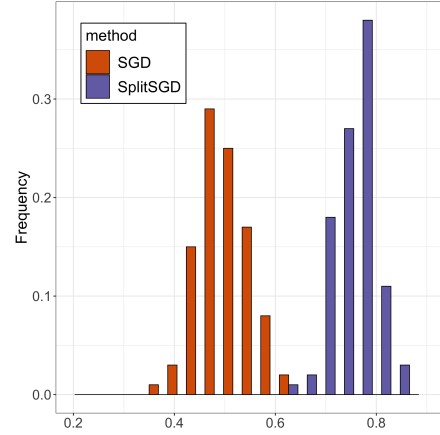

Figure 7: Histogram of the proportion of times that SplitSGD and SGD converged to the global minimum, which shows SplitSGD achieving the global minimum significantly more often. We run 100 simulations and evaluate the proportion 100 times each. We consider 1000 updates with $\eta = 0.01$ and SplitSGD averages its two threads every 200 updates. We consider $a = 2.1$ (left) and $a = 4$ (right).

but also for $\eta = 0.1$ we can see at the very end that the method stabilizes to a higher test accuracy than the competitors.

**Recurrent neural networks (RNNs).** For RNNs, we evaluate a two-layer LSTM (Hochreiter & Schmidhuber, 1997) model on the Penn Treebank (Marcus et al., 1993) language modelling task. We use $\eta \in \{0.1, 0.3, 1.0\}$ for both SGD and SplitSGD, $\eta \in \{0.1, 0.3\}$ for FDR, $\eta \in \{1e-4, 3e-4, 1e-3\}$ for Adam and also introduce SplitAdam, a method similar to SplitSGD, but with Adam in place of SGD with momentum. Here we set the batch size to 20 across all models. As shown in the fourth panel of Figure 6, we can see that SplitSGD outperforms SGD and SplitAdam outperforms Adam with regard to both the best performance and the last performance. FDR is not showing any improvement compared to standard SGD, meaning that in this framework it is unable to detect stationarity and decay the learning rate accordingly. Similar to what already observed with the CNN, we need to note that our proposed splitting strategy has the advantage of reducing the effect of overfitting, which is very severe for SGD, Adam and FDR while very small for SplitAdam and SplitSGD. We postpone the theoretical understanding for this phenomena as our future work, but we make an attempt to develop an intuition on why this could be the case with an example.

We focus on the effect of averaging the two threads, and consider a very simple polynomial loss landscape in dimension 2 with a local minimum and a global minimum. The loss surface is $f(x, y) = (x^2 + y^2 - 1) \cdot (x^2 + y^2 - a(x + y) + 1)$ and the parameter $a$ regulates the distance and the difference in depth of the two minima. When $a = 2$ both minima are global, when $a > 2$ the minimum located in the positive quadrant becomes the only global one and its basin of attraction gets larger. The histograms in Figure 7 refer to the proportion of times that the convergence happened to the global minimum instead of the local one when starting exactly on the saddle point located between them. In the left panel we set $a = 2.1$, while in the right we set $a = 4$, and we see that as $a$ grows the advantage of averaging the two threads becomes more relevant, since the new starting point falls into the basin of attraction of the global minimum more often. This example is not intended to completely explain the difference in overfitting between SplitSGD and the other methods that we see in Figure 6, but just to provide an intuition on why the averaging could be beneficial.

For the deep neural networks considered here, SplitSGD shows better results compared to SGD and Adam, and exhibits strong robustness to the choice of initial learning rates, which further verifies the effectiveness of SplitSGD in deep neural networks. The Splitting Diagnostic is proved to be beneficial in all these different settings, reducing the learning rate to enhance the test performance and reduce overfitting of the networks. FDR shows a good result when used on ResNet with a specific learning rate, but in the other setting is not improving over SGD, suggesting that its diagnostic does not work on a variety of different scenarios.

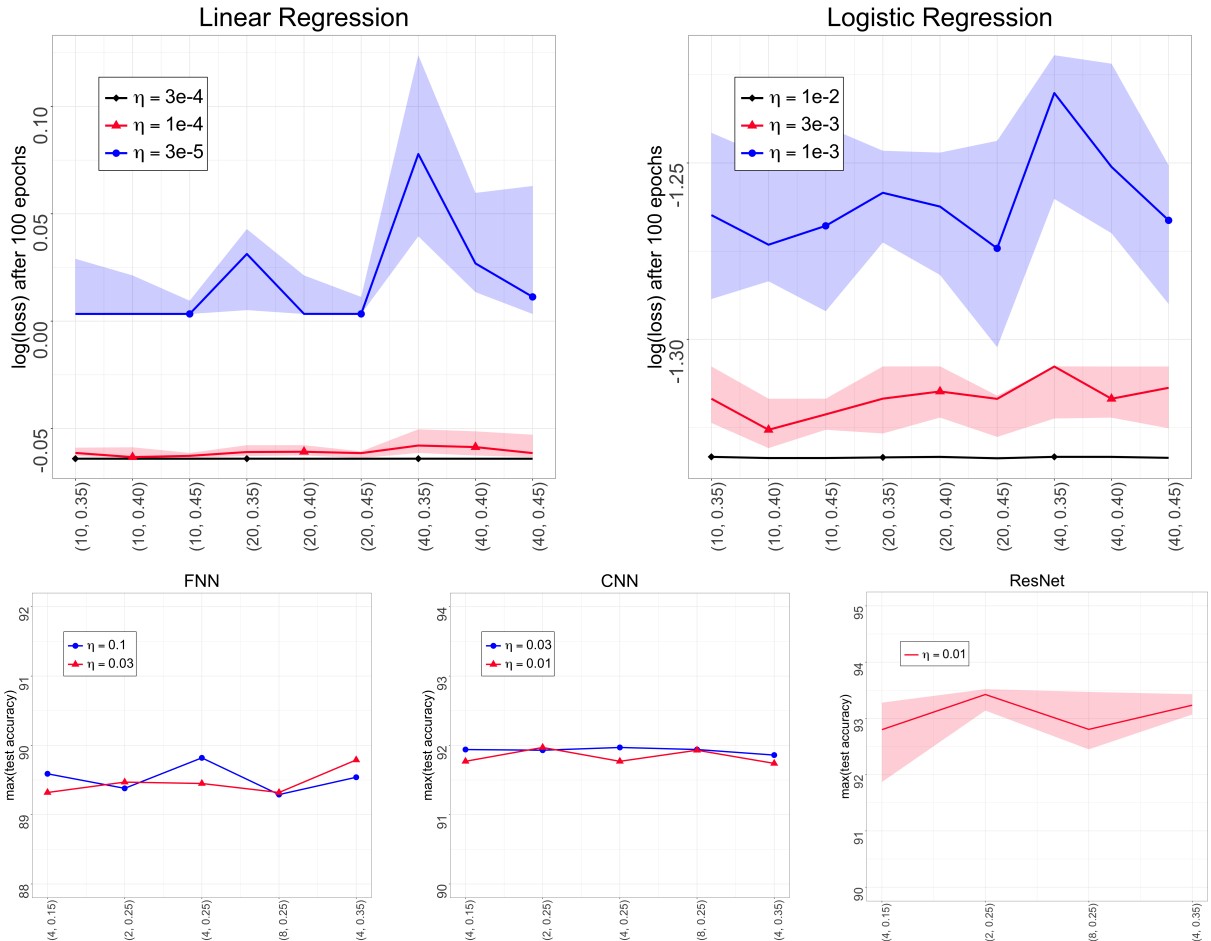

Figure 8: Sensitivity analysis for SplitSGD with respect to the parameters $w$ and $q$, appearing as the labels of the x-axis in the form $(w, q)$. In the convex setting (top plots) we consider the log loss achieved after 100 epochs, while for deep neural networks (bottom plots) we report the maximum of the test accuracy in 100 epochs. For ResNet, we add a confidence band to highlight the stability of the results over different runs. Overall, SplitSGD achieves stable performance across various hyper-parameters setups; for details, see Section 4.3.

### 4.3 Sensitivity Analysis for SplitSGD

In this section, we analyse the impact of the hyper-parameters in the SplitSGD procedure. We focus on $q$ and $w$, while $l$ changes so that the computational budget of each diagnostic is fixed at one epoch. In the top panels of Figure 8 we analyse the sensitivity of SplitSGD to these two parameters in the convex setting, for both linear and logistic regression, and consider $w \in \{10, 20, 40\}$ and $q \in \{0.35, 0.40, 0.45\}$. The data are generated in the same way as those used in Section 4.1. On the y-axis we report the log(loss) after training for 100 epochs, while on the x-axis we consider the different $(w, q)$ configurations. The results are as expected; when the initial learning rate is larger, the impact of these parameters is very modest. When the initial learning rate is small, having a quicker decay (i.e. setting $q$ smaller) worsen the performance.

In the bottom panels of Figure 8 we see the same analysis applied to the FeedForward Neural Network (FNN) described in Appendix D and the CNN used before, both trained on Fashion-MNIST. Here we report the maximum test accuracy achieved when training for 100 epochs, and on the x-axis we have various configurations for $q \in \{0.15, 0, 25, 0.35\}$ and $w \in \{2, 4, 8\}$. The results are very encouraging, showing that SplitSGD is robust with respect to the choice of these parameters also in non-convex settings. In the last plot of the bottom row we also run four simulation with each pair of $w$ and $q$ that are different from the

default values used in Figure 6. For each run we compute the maximum test accuracy over 350 epochs and for each $(w, q)$ we report both the median (solid line) and a shaded region corresponding to the minimum and maximum over the four experiments. What we observe is that the variability is in general pretty small, confirming that the goodness of the SplitSGD performance is not to be attributed to a fine tuning of these two parameters.

## 5 Conclusion and Future Work

We have developed a novel optimization method called SplitSGD, that works by splitting the SGD thread for stationarity detection. Extensive simulation studies show that this method is robust to the choice of the initial learning rate in a variety of optimization tasks, compared to classic adaptive and non-adaptive methods. Moreover, SplitSGD on certain deep neural network architectures outperforms classic SGD, Adam and FDR in terms of the test accuracy, and can sometime limit greatly the impact of overfitting. As the critical element underlying SplitSGD, the Splitting Diagnostic is a simple yet effective strategy that can possibly be incorporated into many optimization methods beyond SGD, as we already showed training SplitAdam on LSTM. One possible limitation of this method is the introduction of a new relevant parameter $q$, that regulates the rate at which the learning rate is adaptively decreased. Our simulations suggest the use of two different values depending on the context. A slower decrease, $q = 0.4$, in convex optimization, and a more aggressive one, $q = 0.25$, for deep learning. In the future, we look forward to seeing research investigations toward boosting the convergence of SplitSGD by allowing for different learning rate selection strategies across different layers of the neural networks.

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

# A    Description of the convex setting and choice of the tolerance parameter $q$

For the experiments in the convex setting we use a feature matrix $X \in \mathbb{R}^{n \times d}$ with standard normal entries and $n = 1000$, $d = 20$. We set $\theta_j^* = 5 \cdot e^{-j/2}$ for $j = 1, ..., 20$ to guarantee some difference in the entries. We generate the linear data as $y_i = X_i \cdot \theta^* + \epsilon_i$, where $\epsilon_i \sim N(0, 1)$, and the data for logistic regression from a Bernoulli with probability $(1 + e^{-X_i \cdot \theta^*})^{-1}$. The other parameters that are used through all Section 4.1 are the numbers of windows $w = 20$ of size $l = 50$ (so that each diagnostic consists of one epoch), the length of the first single thread $t_1 = 4$ epochs, and the acceptance proportion $q = 0.4$.

As we say in the main text, in general we would like $w, l, t_1$ and the number of diagnostics $B$ to be as large as possible, given the computational budget that we have. The tolerance $q$, instead, is more tricky. In Theorem 3 and Figure 3 we shown that, as $t_1 \to \infty$, the distribution of the sign of the gradient coherence is approximately a coin flip, provided that $\eta$ is small enough. This means that, once stationarity is reached, we want $q$ not to be too big, so that we will not observe a proportion of negative gradient coherences smaller than $q$ just by chance too often (and erroneously think that stationarity has not been reached yet). If we were then to assume independence between the $Q_i$, we should set $q$ to control the probability of a type I error (returning $T_D = N$ even though stationarity has been reached), which is

$$\frac{1}{2^w} \sum_{i=0}^{\lfloor w \cdot q \rfloor - 1} \binom{w}{i}$$

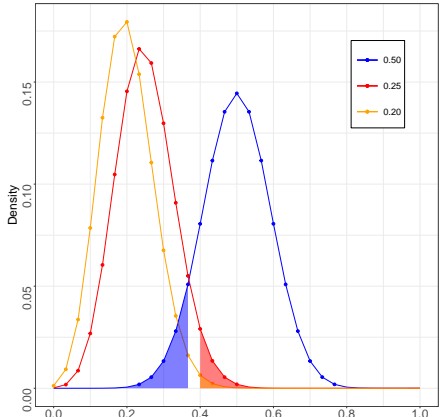 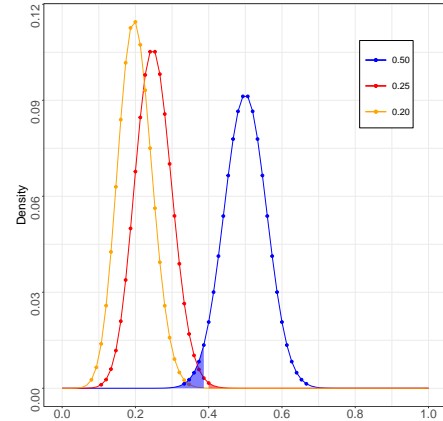

Figure 9: Continuous representation of the probability mass function of Binomial distributions. On the left we set $w = 30$ and $q = 0.4$, on the right $w = 75$ and $q = 0.4$, for both the probability of success (observing a negative gradient coherence) is $p \in \{0.2, 0.25, 0.5\}$. When $p = 0.5$ (stationarity) the type I error happens with probability approximated by the shaded blue region. When $p < 0.5$ (non stationarity) we erroneously declare stationarity with probability approximated by the shaded red and orange region.

However, if we set $q$ to be too small, then in the initial phases of the procedure we might think that we have already reached stationarity only because by chance we observed a proportion of negative dot products larger than $q$. This trade-off, represented in Figure 9, is particularly relevant if we cannot afford a large number of windows $w$, but it loses importance as $w$ grows.

## B  Comparison with `pflug` Diagnostic with different parameters

In Figure 10 and Figure 11 we see other configurations for the experiment reported in the top panels of Figure 5. There, the starting point was set to be around $\theta_s$, where $\theta_{s,j} = 5 \cdot e^{-(d-j)/2}$ for $j = 1, ..., 20$. Here we consider the same starting point for the panels on the right (for both linear and logistic regression) but a smaller learning rate. In both cases it is extremely clear that the `pflug` Diagnostic is detecting stationarity too late, and often (in the case of linear regression) running to the end of the budget. This can be a big problem in practice, because after stationarity has been reached all the iterations that keep using the same learning rate are not going to improve convergence, and are fundamentally wasted. In the left and middle panel of both figures we consider a starting point for the procedures around the minimizer $\theta^*$. In this scenario, for both larger and smaller learning rates, we see that both procedure are either very precise or detect stationarity a bit too early. This is a smaller problem in practice, since at that point the learning rate is reduced but the SGD procedures keep running, even if with a smaller learning rate. The speed of convergence is then slower, but the steps that we make are still important towards convergence.

## C  Changes to the SplitSGD procedure in deep learning

The differences between the SplitSGD procedure that we analysed in Section 2 and its adaptation to deep learning are the following:

- **momentum of SGD:** while in the convex setting we study the behavior of vanilla SGD, when training deep neural networks the standard choice is to use SGD with momentum (Sutskever et al., 2013), which updates as

$$
\begin{aligned}
\Delta\theta_t &= \beta \cdot \Delta\theta_{t-1} - \eta_t \cdot g(\theta_t, Z_{t+1}) \\
\theta_{t+1} &= \theta_t + \Delta\theta_t
\end{aligned}
\tag{12}
$$

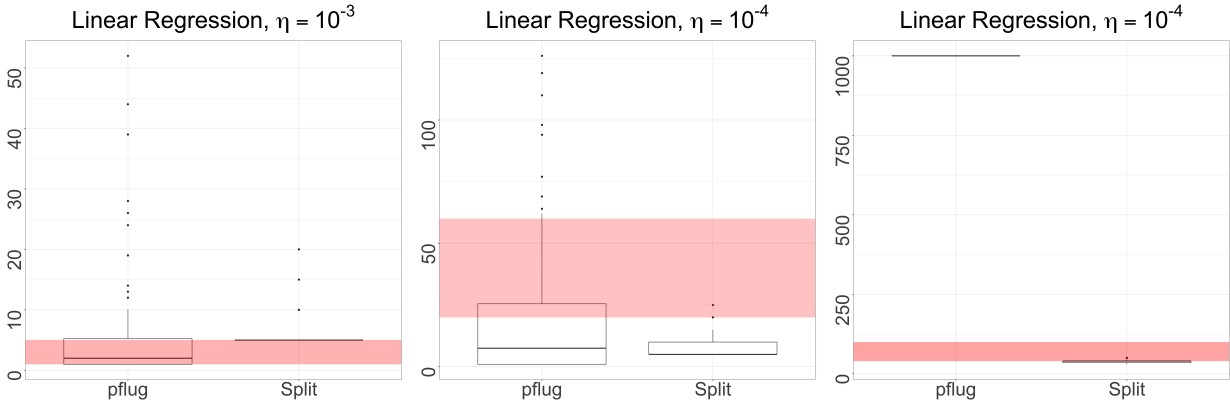

Figure 10: (left) starting around $\theta^*$, large learning rate. (middle) starting around $\theta^*$, small learning rate. (right) starting around $\theta_s$, small learning rate.

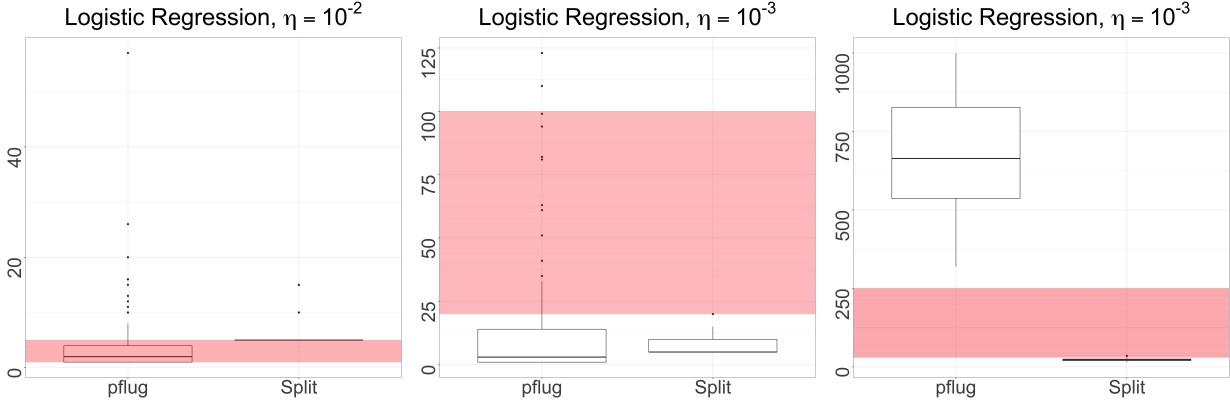

Figure 11: (left) starting around $\theta^*$, large learning rate. (middle) starting around $\theta^*$, small learning rate. (right) starting around $\theta_s$, small learning rate.

If the learning rate is kept constant, SGD with momentum still goes through a transient phase before reaching stationarity. Even in this case, as we already saw in (3), we have that $\mathbb{E}_{\theta \sim \pi_{\eta}}[g(\theta, Z)] = 0$ since we see from (12) that $\mathbb{E}_{\theta \sim \pi_{\eta}}[\Delta\theta] = 0$. This justifies the use of the gradient coherence as defined in (6) also when considering SGD with momentum.

- **gradient coherence on layers:** when considering a parameter space of dimension $d$, the gradient coherence is a dot product of two $d$-dimensional vectors. In deep learning, the parameter space is usually extremely large, so we decided to divide these vectors into pieces to try to extract more information about the stationarity of the SGD updates, by computing the dot product of each of the pieces. In practice, let's divide the vectors $u$ and $v$ into $p$ pieces not necessarily of equal length, so that $v = (v_1, v_2, ..., v_p)$ and $u = (u_1, u_2, ..., u_p)$. Instead of computing the single dot product $\langle v, u \rangle$ we store the $p$ dot products $\langle v_i, u_i \rangle$. In this way, we can also relax the trade-off between $l$ and $w$ (remember that we want to allocate a single epoch to the diagnostic, so that $2lw$ is fixed to be the size of the training set). By computing more than a single value of $Q$ for each pair of vectors, we can allow to set $w$ smaller.

  A natural division of the parameter space into smaller pieces comes from the layers of the network, so each time we compute the gradient coherence of the two threads we actually compute a separate value for each layer and then store all of them together. In the final count, as we did in the non-convex setting, we look at the proportion of these values that are negative to decide whether to decay the learning rate.

- **length of the single thread:** since training deep neural networks is usually computationally expensive, we decided not to increase the length of the single thread after stationarity was detected. This is made simply to avoid situations where stationarity is detected early and the length of the single thread increases so fast that we do not have time to decay the learning rate by much before reaching the end of the computational budget that we allocated.

- **hyperparameters for the diagnostic:** we set the relevant hyperparameters $w$ and $q$ to take value $w = 4$ and $q = 0.25$. The value of $w$ is much smaller than the one used in the convex setting for the reason explained above that we compute the gradient coherence separately for each layer. With this choice, we can dedicate $1/8$ of the updates of each epoch to compute for each thread the average of the gradients and be sure that we averaged out a lot of the noise. The choice of setting $q = 0.25$ comes from the empirical results that we observed, and a deeper study of this parameter is probably needed. We performed a sensitivity analysis on these two parameters in Section 4.3 and noticed that a departure from these values is not changing the performance by much.

## D   Other experiments in deep learning

We add here the description of two more experiments in Deep Learning that did not fit in the main body of the paper, together with the plots that we already included in Figure 6 but this time with the addition of the 90% confidence bands. We see that the Splitting Diagnostic increases the variability of SplitSGD with respect to other methods in some settings, but the interpretation that we gave in Section 4.2 of the better performance of SplitSGD and the lack of overfitting holds.

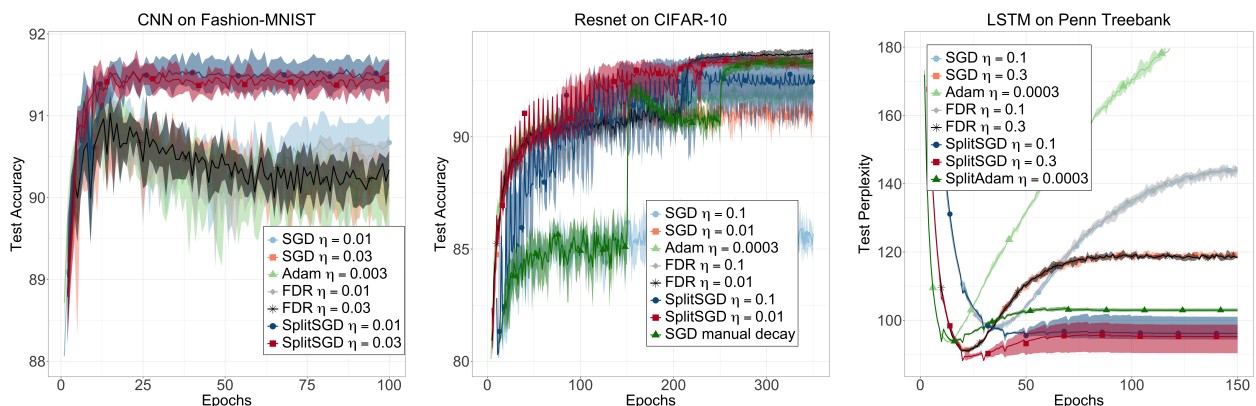

Figure 12: This is the same as Figure 6 but here we also added 90% confidence bands for each method.

**Feedforward neural networks (FNNs).** We train a FNN with three hidden layers of size $256, 128$ and $64$ on the Fashion-MNIST dataset (Xiao et al., 2017). The network is fully connected, with ReLu activation functions. The initial learning rates are $\eta \in \{1e-2, 3e-2, 1e-1\}$ for SGD and SplitSGD and $\eta \in \{3e-4, 1e-3, 3e-3\}$ for Adam. In the first panel of Figure 13 we see that most methods achieve very good accuracy, but SplitSGD reaches the overall best test accuracy when $\eta = 1e-1$ and great accuracy with small oscillations when $\eta = 3e-2$. The peaks in the SplitSGD performance are usually due to the averaging, while the smaller oscillations are due to the learning rate decay.

**VGG19.** When training the neural network VGG19[3] on CIFAR-10, we observe a similar behavior to what already shown when training ResNet (second panel of Figure 6). SplitSGD, with both learning rates $1e-1$ and $1e-2$ achieves the same test accuracy of the manually tuned SGD, but in less epochs, and beats the performance of SGD and Adam. Also here it is possible to see the spikes given by the averaging, followed by the smoothing caused by the learning rate decay.

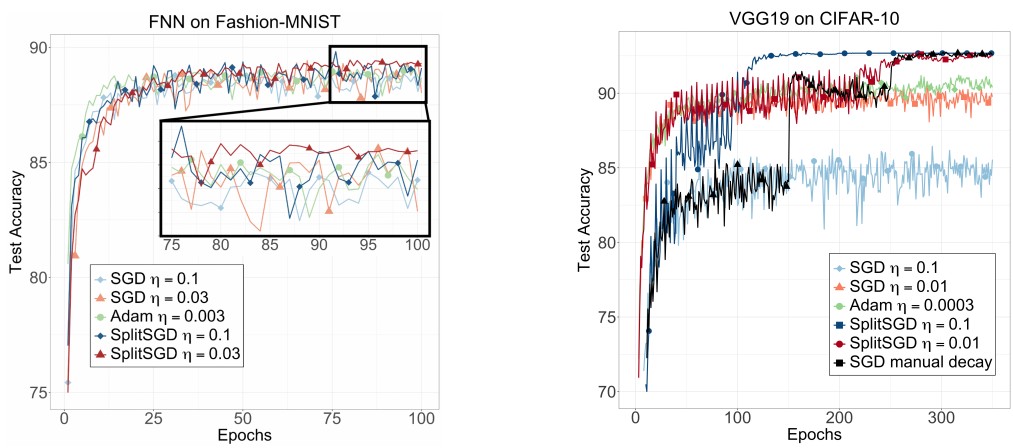

Figure 13: Compare the accuracy of SGD, Adam and SplitSGD in training FNN on Fashion-MNIST (left) and VGG19 on CIFAR-10 (right).

# E   Lemmas

Proof of Lemma 4:

---

[3]More details can be found in `https://pytorch.org/docs/stable/torchvision/models.html`

**Proof** This proof can be easily adapted from Moulines & Bach (2011). From the recursive definition of $\theta_t$ one has

$$\mathbb{E}\left[||\theta_t - \theta^*||^2\right] \leq \left(1 - 2\eta(\mu - L^2\eta)\right) \cdot \mathbb{E}\left[||\theta_{t-1} - \theta^*||^2\right] + 2G^2\eta^2.$$

This inequality can be recursively applied to obtain the desired result

$$\mathbb{E}\left[||\theta_t - \theta^*||^2\right] \leq \left(1 - 2\eta(\mu - L^2\eta)\right)^t \cdot \mathbb{E}\left[||\theta_0 - \theta^*||^2\right] + 2G^2\eta^2 \sum_{j=0}^{t-1} \left(1 - 2\eta(\mu - L^2\eta)\right)^j$$

$$\leq \left(1 - 2\eta(\mu - L^2\eta)\right)^t \cdot \mathbb{E}\left[||\theta_0 - \theta^*||^2\right] + \frac{G^2\eta}{\mu - L^2\eta}$$

∎

This lemma represents the dynamic of SGD with constant learning rate, where the dependence from the starting point vanishes exponentially fast, but there is a term dependent on $\eta$ that is not vanishing even for large $t$.

**Lemma 6** *If Assumption 3.4 with $m = 4$ holds, then for any $t, i \in \mathbb{N}$ one has*

$$\mathbb{E}\left[||\theta_{t+i} - \theta_t||^4 \mid \mathcal{F}_t\right] \leq \eta^4 i^4 G^4$$

**Proof** For any $j = 1, ..., l$, let $x_j$ be a vector of length $n$. Applying Cauchy-Schwarz inequality twice, we get

$$||\sum_{j=1}^{l} x_j||^4 = ||\sum_{j=1}^{l} x_j||^2 \cdot ||\sum_{j=1}^{l} x_j||^2 \leq \left(l \cdot \sum_{j=1}^{l} ||x_j||^2\right)^2$$

$$= l^2 \left(\sum_{j=1}^{l} ||x_j||^2\right)^2 \leq l^3 \cdot \sum_{j=1}^{l} ||x_j||^4 \tag{13}$$

Since

$$\theta_{t+i} = \theta_t - \eta \sum_{j=0}^{i-1} g(\theta_{t+j}, Z_{t+j+1}),$$

then we can use the fact that $\mathcal{F}_k \subseteq \mathcal{F}_{k+1}$ for any $k$, together with Assumption 3.4 and (13), to get that

$$\mathbb{E}\left[||\theta_{t+i} - \theta_t||^4 \mid \mathcal{F}_t\right] = \eta^4 \cdot \mathbb{E}\left[||\sum_{j=0}^{i-1} g(\theta_{t+j}, Z_{t+j+1})||^4 \mid \mathcal{F}_t\right]$$

$$\leq \eta^4 i^3 \sum_{j=0}^{i-1} \mathbb{E}\left[||g(\theta_{t+j}, Z_{t+j+1})||^4 \mid \mathcal{F}_t\right]$$

$$= \eta^4 i^3 \sum_{j=0}^{i-1} \mathbb{E}\big[\underbrace{\mathbb{E}\left[||g(\theta_{t+j}, Z_{t+j+1})||^4 \mid \mathcal{F}_{t+j}\right]}_{\leq G^4} \mid \mathcal{F}_t\big]$$

$$\leq \eta^4 i^4 G^4$$

Note that this is a bound that considers the worst case in which all the noisy gradient updates point in the same direction and are of norm $G$.

∎

**Remark 7** *We can obviously use the same bound for the unconditional squared norm, since*

$$\mathbb{E}\left[||\theta_{t+i} - \theta_t||^4\right] = \mathbb{E}\left[\mathbb{E}\left[||\theta_{t+i} - \theta_t||^4 \mid \mathcal{F}_t\right]\right] \leq \eta^4 i^4 G^4.$$

**Lemma 8** *If Assumption 3.2 and 3.4 with $m = 2$ hold, then for any $i = 1, ..., l$ and $k = 1, 2$ we have that*

$$\mathbb{E}\left[||\nabla F(\theta_{t+i}^{(k)}) - \nabla F(\theta_0)||^2 \mid \mathcal{F}_t\right] \leq (L||\theta_t - \theta_0|| + L\eta Gi)^2$$

**Proof** By adding and subtracting $\nabla F(\theta_t)$, and by Lemma 6, we get.

$$\mathbb{E}\left[||\nabla F(\theta_{t+i}^{(k)}) - \nabla F(\theta_0)||^2 \mid \mathcal{F}_t\right] \leq \mathbb{E}\left[||\nabla F(\theta_{t+i}^{(k)}) - \nabla F(\theta_t) + \nabla F(\theta_t) - \nabla F(\theta_0)||^2 \mid \mathcal{F}_t\right]$$

$$\leq ||\nabla F(\theta_t) - \nabla F(\theta_0)||^2 + \mathbb{E}\left[||\nabla F(\theta_{t+i}^{(k)}) - \nabla F(\theta_t)||^2 \mid \mathcal{F}_t\right]$$

$$+ 2||\nabla F(\theta_t) - \nabla F(\theta_0)|| \cdot \mathbb{E}\left[||\nabla F(\theta_{t+i}^{(k)}) - \nabla F(\theta_t)|| \mid \mathcal{F}_t\right]$$

$$\leq L^2||\theta_t - \theta_0||^2 + L^2\mathbb{E}\left[||\theta_{t+i}^{(k)} - \theta_t||^2 \mid \mathcal{F}_t\right] + 2L^2||\theta_t - \theta_0|| \cdot \mathbb{E}\left[||\theta_{t+i}^{(k)} - \theta_t|| \mid \mathcal{F}_t\right]$$

$$\leq L^2||\theta_t - \theta_0||^2 + L^2\eta^2 G^2 i^2 + 2L^2||\theta_t - \theta_0||\eta Gi$$

$$= (L||\theta_t - \theta_0|| + L\eta Gi)^2$$

∎

**Remark 9** *When we consider the unconditional distance of the gradients, we can simply use smoothness and Remark 7 to get*

$$\mathbb{E}\left[||\nabla F(\theta_{t+i}^{(k)}) - \nabla F(\theta_0)||^2\right] \leq L^2\mathbb{E}\left[||\theta_{t+i}^{(k)} - \theta_0||^2\right] \leq L^2\eta^2 G^2 (t+i)^2$$

*which is the same result that we obtain from Lemma 8 if at the end we bound $||\theta_t - \theta_0||$ with its expectation, and use the fact that $\mathbb{E}[||\theta_t - \theta_0||] \leq \eta Gt$.*

**Lemma 10** *If Assumption 3.2 and 3.4 with $m = 2$ hold, then for any $i = 1, ..., l$ and $k = 1, 2$ we have that*

*i)* $\mathbb{E}\left[||\nabla F(\theta_{t+i}^{(k)})||^2 \mid \mathcal{F}_t\right] \leq (||\nabla F(\theta_0)|| + L||\theta_t - \theta_0|| + L\eta Gi)^2$

*ii)* $\mathbb{E}\left[||\nabla F(\theta_{t+i}^{(k)})||^2 \mid \mathcal{F}_t\right] \leq (L||\theta_t - \theta^*|| + L\eta Gi)^2$

**Proof** We add and subtract $\nabla F(\theta_t)$ to the gradient on the left hand side, and apply Lemma 6.

$$\mathbb{E}\left[||\nabla F(\theta_{t+i}^{(k)})||^2 \mid \mathcal{F}_t\right] = \mathbb{E}\left[||\nabla F(\theta_{t+i}^{(k)}) - \nabla F(\theta_t) + \nabla F(\theta_t)||^2 \mid \mathcal{F}_t\right]$$

$$\leq ||\nabla F(\theta_t)||^2 + \mathbb{E}\left[||\nabla F(\theta_{t+i}^{(k)}) - \nabla F(\theta_t)||^2 \mid \mathcal{F}_t\right]$$

$$+ 2||\nabla F(\theta_t)|| \cdot \mathbb{E}\left[||\nabla F(\theta_{t+i}^{(k)}) - \nabla F(\theta_t)|| \mid \mathcal{F}_t\right]$$

$$\leq ||\nabla F(\theta_t)||^2 + L^2\mathbb{E}\left[||\theta_{t+i}^{(k)} - \theta_t||^2 \mid \mathcal{F}_t\right] + 2L||\nabla F(\theta_t)|| \cdot \mathbb{E}\left[||\theta_{t+i}^{(k)} - \theta_t|| \mid \mathcal{F}_t\right]$$

$$\leq ||\nabla F(\theta_t)||^2 + L^2\eta^2 G^2 i^2 + 2||\nabla F(\theta_t)|| \cdot L\eta Gi \tag{14}$$

To get part *i)* we repeat the same trick, this time adding and subtracting $\nabla F(\theta_0)$ to the terms that contain $\nabla F(\theta_t)$.

$$(14) \leq ||\nabla F(\theta_0)||^2 + ||\nabla F(\theta_t) - \nabla F(\theta_0)||^2 + 2||\nabla F(\theta_0)|| \cdot ||\nabla F(\theta_t) - \nabla F(\theta_0)||$$

$$+ L^2\eta^2 G^2 i^2 + 2||\nabla F(\theta_0)|| \cdot L\eta Gi + 2||\nabla F(\theta_t) - \nabla F(\theta_0)|| \cdot L\eta Gi$$

$$\leq ||\nabla F(\theta_0)||^2 + L^2||\theta_t - \theta_0||^2 + 2L||\nabla F(\theta_0)|| \cdot ||\theta_t - \theta_0||$$

$$+ L^2\eta^2 G^2 i^2 + 2||\nabla F(\theta_0)|| \cdot L\eta Gi + 2||\theta_t - \theta_0|| \cdot L^2\eta Gi$$

$$= (||\nabla F(\theta_0)|| + L||\theta_t - \theta_0|| + L\eta Gi)^2$$

To get part $ii)$, instead, we can add $\nabla f(\theta^*)$ and get

$$
\begin{aligned}
(14) &\leq L^2||\theta_t - \theta^*||^2 + L^2\eta^2 G^2 i^2 + 2||\theta_t - \theta^*|| \cdot L^2\eta Gi \\
&= (L||\theta_t - \theta^*|| + L\eta Gi)^2
\end{aligned}
$$

$\blacksquare$

**Remark 11** *For the unconditional squared norm of the gradient we again obtain the same bound as if in Lemma 10 we were considering $\mathbb{E}[||\theta_t - \theta_0||] \leq \eta Gt$ instead of just the argument of the expectation.*

$$
\begin{aligned}
\mathbb{E}\left[||\nabla F(\theta_{t+i}^{(k)})||^2\right] &= \mathbb{E}\left[||\nabla F(\theta_{t+i}^{(k)}) - \nabla F(\theta_0) + \nabla F(\theta_0)||^2\right] \\
&\leq ||\nabla F(\theta_0)||^2 + \mathbb{E}\left[||\nabla F(\theta_{t+i}^{(k)}) - \nabla F(\theta_0)||^2\right] \\
&\quad + 2||\nabla F(\theta_0)|| \cdot \mathbb{E}\left[||\nabla F(\theta_{t+i}^{(k)}) - \nabla F(\theta_0)||\right] \\
&\leq ||\nabla F(\theta_0)||^2 + L^2\eta^2 G^2(t+i)^2 + 2||\nabla F(\theta_0)||L\eta G(t+i) \\
&= (||\nabla F(\theta_0)|| + L\eta G(t+i))^2
\end{aligned}
$$

# F  Proof of Theorem 2

To slightly simplify the notation, we consider only $Q_1$. For the following windows, the calculations are equal and just involve some more terms, that are negligible if $\eta$ is small enough. We assume that the Splitting Diagnostic starts after $t$ iterations have already been made. We use the idea that, for a fixed $t$, if the learning rate is sufficiently small, the SGD iterate $\theta_t$ and $\theta_0$ will not be very far apart. In particular we will use $\eta$ small enough such that $\eta \cdot (t+l)$ is small, making every term of order $O(\eta^k(t+l)^k)$ negligible for $k > 1$. Thanks to the conditional independence of the errors, the expectation of $Q_1$ can be written only in terms of the true gradients.

$$
\begin{aligned}
\mathbb{E}[Q_1] &= \frac{1}{l^2}\sum_{i=0}^{l-1}\sum_{j=0}^{l-1}\mathbb{E}\left[\langle g(\theta_{t+i}^{(1)}), g(\theta_{t+j}^{(2)})\rangle\right] \\
&= \frac{1}{l^2}\sum_{i=0}^{l-1}\sum_{j=0}^{l-1}\mathbb{E}\left[\langle \nabla F(\theta_{t+i}^{(1)}) + \epsilon(\theta_{t+i}^{(1)}), \nabla F(\theta_{t+j}^{(2)}) + \epsilon(\theta_{t+j}^{(2)})\rangle\right] \\
&= \frac{1}{l^2}\sum_{i=0}^{l-1}\sum_{j=0}^{l-1}\mathbb{E}\left[\langle \nabla F(\theta_{t+i}^{(1)}), \nabla F(\theta_{t+j}^{(2)})\rangle\right] \qquad (15)
\end{aligned}
$$

We now add and subtract $\nabla F(\theta_0)$, and use L-smoothness and Remark 7 to provide a lower bound for $\mathbb{E}\left[Q_1\right]$. From (15) we get

$$\mathbb{E}\left[Q_1\right] = \frac{1}{l^2}\sum_{i=0}^{l-1}\sum_{j=0}^{l-1}\left\{\langle\nabla F(\theta_0),\nabla F(\theta_0)\rangle + \mathbb{E}\left[\langle\nabla F(\theta_{t+i}^{(1)}) - \nabla F(\theta_0),\nabla F(\theta_{t+j}^{(2)}) - \nabla F(\theta_0)\rangle\right]\right.$$

$$\left. + \mathbb{E}\left[\langle\nabla F(\theta_0),\nabla F(\theta_{t+j}^{(2)}) - \nabla F(\theta_0)\rangle\right] + \mathbb{E}\left[\langle\nabla F(\theta_{t+i}^{(1)}) - \nabla F(\theta_0),\nabla F(\theta_0)\rangle\right]\right\}$$

$$\geq ||\nabla F(\theta_0)||^2 - \frac{1}{l^2}\sum_{i=0}^{l-1}\sum_{j=0}^{l-1}\mathbb{E}\left[||\nabla F(\theta_{t+i}^{(1)}) - \nabla F(\theta_0)|| \cdot ||\nabla F(\theta_{t+j}^{(2)}) - \nabla F(\theta_0)||\right]$$

$$- \frac{1}{l}\sum_{j=0}^{l-1}\mathbb{E}\left[||\nabla F(\theta_0)|| \cdot ||\nabla F(\theta_{t+j}^{(2)}) - \nabla F(\theta_0)||\right]$$

$$- \frac{1}{l}\sum_{i=0}^{l-1}\mathbb{E}\left[||\nabla F(\theta_0)|| \cdot ||\nabla F(\theta_{t+i}^{(1)}) - \nabla F(\theta_0)||\right] \tag{16}$$

$$\geq ||\nabla F(\theta_0)||^2 - \frac{L^2}{l^2}\sum_{i=0}^{l-1}\sum_{j=0}^{l-1}\sqrt{\mathbb{E}\left[||\theta_{t+i}^{(1)} - \theta_0||^2\right] \cdot \mathbb{E}\left[||\theta_{t+j}^{(2)} - \theta_0||^2\right]}$$

$$- \frac{2L}{l}\sum_{i=0}^{l-1}||\nabla F(\theta_0)|| \cdot \mathbb{E}\left[||\theta_{t+i}^{(1)} - \theta_0||\right]$$

$$\geq ||\nabla F(\theta_0)||^2 - L^2\eta^2 G^2(t+l)^2 - 2L||\nabla F(\theta_0)||\eta G(t+l)$$

$$= ||\nabla F(\theta_0)||^2 - 2L||\nabla F(\theta_0)||\eta G(t+l) + O(\eta^2(t+l)^2) \tag{17}$$

Notice that, in the extreme case where $\eta = 0$, we simply have $\mathbb{E}[Q_1] \geq ||\nabla F(\theta_0)||^2$ which is actually an equality, since we would have $\theta_t = \theta_0$ and the noisy gradient at step $t$ would be $g(\theta_0, Z_t)$, whose expectation

is just $\nabla F(\theta_0)$. We now expand the second moment, and there are a lot of terms to be considered separately.

$$
\begin{aligned}
l^4 \cdot \mathbb{E}\left[Q_1^2\right] &= \mathbb{E}\left[\left\langle \sum_{i=0}^{l-1} g(\theta_{t+i}^{(1)}), \sum_{j=0}^{l-1} g(\theta_{t+j}^{(2)})\right\rangle^2\right] \\
&= \mathbb{E}\left[\left\langle \sum_{i=0}^{l-1} \left(\nabla F(\theta_{t+i}^{(1)}) + \epsilon(\theta_{t+i}^{(1)})\right), \sum_{j=0}^{l-1} \left(\nabla F(\theta_{t+j}^{(2)}) + \epsilon(\theta_{t+j}^{(2)})\right)\right\rangle^2\right] \\
&= \underbrace{\mathbb{E}\left[\left\langle \sum_{i=0}^{l-1} \nabla F(\theta_{t+i}^{(1)}), \sum_{j=0}^{l-1} \nabla F(\theta_{t+j}^{(2)})\right\rangle^2\right]}_{I} + \underbrace{\mathbb{E}\left[\left\langle \sum_{i=0}^{l-1} \nabla F(\theta_{t+i}^{(1)}), \sum_{j=0}^{l-1} \epsilon(\theta_{t+j}^{(2)})\right\rangle^2\right]}_{II} \\
&\quad + \underbrace{\mathbb{E}\left[\left\langle \sum_{i=0}^{l-1} \epsilon(\theta_{t+i}^{(1)}), \sum_{j=0}^{l-1} \nabla F(\theta_{t+j}^{(2)})\right\rangle^2\right]}_{III} + \underbrace{\mathbb{E}\left[\left\langle \sum_{i=0}^{l-1} \epsilon(\theta_{t+i}^{(1)}), \sum_{j=0}^{l-1} \epsilon(\theta_{t+j}^{(2)})\right\rangle^2\right]}_{IV} \\
&\quad + 2\underbrace{\mathbb{E}\left[\left\langle \sum_{i=0}^{l-1} \nabla F(\theta_{t+i}^{(1)}), \sum_{j=0}^{l-1} \nabla F(\theta_{t+j}^{(2)})\right\rangle \cdot \left\langle \sum_{h=0}^{l-1} \nabla F(\theta_{t+h}^{(1)}), \sum_{k=0}^{l-1} \epsilon(\theta_{t+k}^{(2)})\right\rangle\right]}_{V} \\
&\quad + 2\underbrace{\mathbb{E}\left[\left\langle \sum_{i=0}^{l-1} \nabla F(\theta_{t+i}^{(1)}), \sum_{j=0}^{l-1} \nabla F(\theta_{t+j}^{(2)})\right\rangle \cdot \left\langle \sum_{h=0}^{l-1} \epsilon(\theta_{t+h}^{(1)}), \sum_{k=0}^{l-1} \nabla F(\theta_{t+k}^{(2)})\right\rangle\right]}_{VI} \\
&\quad + 2\underbrace{\mathbb{E}\left[\left\langle \sum_{i=0}^{l-1} \nabla F(\theta_{t+i}^{(1)}), \sum_{j=0}^{l-1} \nabla F(\theta_{t+j}^{(2)})\right\rangle \cdot \left\langle \sum_{h=0}^{l-1} \epsilon(\theta_{t+h}^{(1)}), \sum_{k=0}^{l-1} \epsilon(\theta_{t+k}^{(2)})\right\rangle\right]}_{VII} \\
&\quad + 2\underbrace{\mathbb{E}\left[\left\langle \sum_{i=0}^{l-1} \nabla F(\theta_{t+i}^{(1)}), \sum_{j=0}^{l-1} \epsilon(\theta_{t+j}^{(2)})\right\rangle \cdot \left\langle \sum_{h=0}^{l-1} \epsilon(\theta_{t+h}^{(1)}), \sum_{k=0}^{l-1} \nabla F(\theta_{t+k}^{(2)})\right\rangle\right]}_{VIII} \\
&\quad + 2\underbrace{\mathbb{E}\left[\left\langle \sum_{i=0}^{l-1} \nabla F(\theta_{t+i}^{(1)}), \sum_{j=0}^{l-1} \epsilon(\theta_{t+j}^{(2)})\right\rangle \cdot \left\langle \sum_{h=0}^{l-1} \epsilon(\theta_{t+h}^{(1)}), \sum_{k=0}^{l-1} \epsilon(\theta_{t+k}^{(2)})\right\rangle\right]}_{IX} \\
&\quad + 2\underbrace{\mathbb{E}\left[\left\langle \sum_{i=0}^{l-1} \epsilon(\theta_{t+i}^{(1)}), \sum_{j=0}^{l-1} \nabla F(\theta_{t+j}^{(2)})\right\rangle \cdot \left\langle \sum_{h=0}^{l-1} \epsilon(\theta_{t+h}^{(1)}), \sum_{k=0}^{l-1} \epsilon(\theta_{t+k}^{(2)})\right\rangle\right]}_{X}
\end{aligned}
$$

In the squared terms $I$ to $IV$, the errors are independent from the other argument of the dot product, conditional on $\mathcal{F}_t$, since they are evaluated on different threads. However, in the double products ($V$ to $X$), some errors are used to generate the subsequent values of the SGD iterates on the same thread. This means that we cannot just ignore them, but we instead have to carefully find an upper bound for each one.

- In $I$ we use the Cauchy-Schwarz inequality and Lemma 10, after exploiting the independence of the two threads conditional on $\mathcal{F}_t$.

$$
\begin{aligned}
\mathbb{E}\left[\left\langle \sum_{i=0}^{l-1} \nabla F(\theta_{t+i}^{(1)}), \sum_{j=0}^{l-1} \nabla F(\theta_{t+j}^{(2)}) \right\rangle^2\right] &\leq l^4 \cdot \max_{i,j}\ \mathbb{E}\left[\left\langle \nabla F(\theta_{t+i}^{(1)}), \nabla F(\theta_{t+j}^{(2)}) \right\rangle^2\right] \\
&\leq l^4 \cdot \max_{i,j}\ \mathbb{E}\left[\mathbb{E}\left[||\nabla F(\theta_{t+i}^{(1)})||^2 \mid \mathcal{F}_t\right] \cdot \mathbb{E}\left[||\nabla F(\theta_{t+j}^{(2)})||^2 \mid \mathcal{F}_t\right]\right] \\
&\leq l^4 \cdot \mathbb{E}\left[(||\nabla F(\theta_0)|| + L||\theta_t - \theta_0|| + L\eta Gl)^4\right] \\
&\lesssim l^4 \cdot \mathbb{E}\left[||\nabla F(\theta_0)||^4 + 4L||\nabla F(\theta_0)||^3 \cdot ||\theta_t - \theta_0|| + 4||\nabla F(\theta_0)||^3 \cdot L\eta Gl\right] \\
&\quad + l^4 \cdot O(\eta^2(t+l)^2) \\
&\lesssim l^4 \cdot \left(||\nabla F(\theta_0)||^4 + 4L\eta G||\nabla F(\theta_0)||^3(t+l) + O(\eta^2(t+l)^2)\right)
\end{aligned}
$$

In the first approximate inequality denoted by $\lesssim$, we have included most of the terms of the expansion in the $O(\eta^2(t+l)^2)$, even if technically we could have done it only after taking the expected value. Notice that here it was important to have a bound in Remark 7 up to the fourth order.

- Terms $II$ and $III$ are equal, since the two threads are identically distributed, and the errors in one thread are a martingale difference sequence independent from the updates in the other thread. We will use the bound for the error norm

$$
\mathbb{E}\left[||\epsilon_t||^2 \mid \mathcal{F}_t\right] = \mathbb{E}\left[\epsilon_t^T \epsilon_t \mid \mathcal{F}_t\right] = \mathbb{E}\left[\operatorname{tr}(\epsilon_t \epsilon_t^T) \mid \mathcal{F}_t\right] \leq d \cdot \sigma_{max} \tag{18}
$$

which is a consequence of Assumption 3.3, and condition on $\mathcal{F}_t$ to use independence of the errors. In the last line we use Remark 11.

$$
\begin{aligned}
\mathbb{E}\left[\left\langle \sum_{i=0}^{l-1} \nabla F(\theta_{t+i}^{(1)}), \sum_{j=0}^{l-1} \epsilon_{t+j}^{(2)} \right\rangle^2\right] &= \sum_{j=0}^{l-1} \mathbb{E}\left[\left\langle \sum_{i=0}^{l-1} \nabla F(\theta_{t+i}^{(1)}), \epsilon_{t+j}^{(2)} \right\rangle^2\right] \\
&\leq l^2 \max_i \sum_{j=0}^{l-1} \mathbb{E}\left[\left|\left|\nabla F(\theta_{t+i}^{(1)})\right|\right|^2 \cdot ||\epsilon_{t+j}^{(2)}||^2\right] \\
&= l^3 \cdot \max_i\ \mathbb{E}\left[\mathbb{E}\left[||\epsilon_{t+i}^{(2)}||^2 \mid \mathcal{F}_t\right] \cdot \mathbb{E}\left[\left|\left|\nabla F(\theta_{t+i}^{(1)})\right|\right|^2 \mid \mathcal{F}_t\right]\right] \\
&\leq l^3 \cdot d\sigma_{max} \cdot \max_i\ \mathbb{E}\left[||\nabla F(\theta_{t+i}^{(1)})||^2\right] \\
&\lesssim l^3 \cdot d\sigma_{max} \cdot \left(||\nabla f(\theta_0)||^2 + 2||\nabla f(\theta_0)||LG\eta(t+l) + O(\eta^2(t+l)^2)\right)
\end{aligned}
$$

- In $IV$, we use the conditional independence of the two threads, and the fact that the errors are a martingale difference sequence, to cancel out all the cross products. An upper bound is then

$$
\begin{aligned}
\mathbb{E}\left[\left\langle \sum_{i=0}^{l-1} \epsilon_{t+i}^{(1)}, \sum_{j=0}^{l-1} \epsilon_{t+j}^{(2)} \right\rangle^2\right] &= \sum_{i=0}^{l-1}\sum_{j=0}^{l-1} \mathbb{E}\left[\left\langle \epsilon_{t+i}^{(1)}, \epsilon_{t+j}^{(2)} \right\rangle^2\right] \\
&\leq \sum_{i=0}^{l-1}\sum_{j=0}^{l-1} \mathbb{E}\left[||\epsilon_{t+i}^{(1)}||^2 \cdot ||\epsilon_{t+j}^{(2)}||^2\right] \\
&= \sum_{i=0}^{l-1}\sum_{j=0}^{l-1} \mathbb{E}\left[\mathbb{E}\left[||\epsilon_{t+i}^{(1)}||^2 \mid \mathcal{F}_t\right] \cdot \mathbb{E}\left[||\epsilon_{t+j}^{(2)}||^2 \mid \mathcal{F}_t\right]\right] \\
&\leq l^2 d^2 \sigma_{max}^2
\end{aligned}
$$

Now we start dealing with the double products. The problem here is that these terms are not all null, since the errors are used in the subsequent updates in the same thread, and they are then not independent.

- $V$ and $VI$ are distributed in the same way. We can cancel out some terms using the conditional independence given $\mathcal{F}_t$, and use the conditional version of Cauchy-Schwarz inequality separately on the two threads.

$$
\mathbb{E}\left[\left\langle \sum_{i=0}^{l-1} \nabla F(\theta_{t+i}^{(1)}), \sum_{j=0}^{l-1} \nabla F(\theta_{t+j}^{(2)}) \right\rangle \cdot \left\langle \sum_{h=0}^{l-1} \nabla F(\theta_{t+h}^{(1)}), \sum_{k=0}^{l-1} \epsilon(\theta_{t+k}^{(2)}) \right\rangle \right]
$$

$$
= \sum_{i,j,h,k=0}^{l-1} \mathbb{E}\left[\left\langle \nabla F(\theta_{t+i}^{(1)}), \nabla F(\theta_{t+j}^{(2)}) \right\rangle \cdot \left\langle \nabla F(\theta_{t+h}^{(1)}), \epsilon(\theta_{t+k}^{(2)}) \right\rangle \right]
$$

$$
= \sum_{i,j,h,k=0}^{l-1} \mathbb{E}\left[\left\langle \nabla F(\theta_0) + \left(\nabla F(\theta_{t+i}^{(1)}) - \nabla F(\theta_0)\right), \nabla F(\theta_0) + \left(\nabla F(\theta_{t+j}^{(2)}) - \nabla F(\theta_0)\right) \right\rangle \times\right.
$$
$$
\left. \times \left\langle \nabla F(\theta_0) + \left(\nabla F(\theta_{t+h}^{(1)}) - \nabla F(\theta_0)\right), \epsilon(\theta_{t+k}^{(2)}) \right\rangle \right]
$$

$$
= \sum_{i,j,h,k=0}^{l-1} \mathbb{E}\left[\left\langle \nabla F(\theta_0), \nabla F(\theta_{t+j}^{(2)}) - \nabla F(\theta_0) \right\rangle \cdot \left\langle \nabla F(\theta_0), \epsilon(\theta_{t+k}^{(2)}) \right\rangle \right]
$$

$$
+ \sum_{i,j,h,k=0}^{l-1} \mathbb{E}\left[\left\langle \nabla F(\theta_0), \nabla F(\theta_{t+j}^{(2)}) - \nabla F(\theta_0) \right\rangle \cdot \left\langle \nabla F(\theta_{t+h}^{(1)}) - \nabla F(\theta_0), \epsilon(\theta_{t+k}^{(2)}) \right\rangle \right]
$$

$$
+ \sum_{i,j,h,k=0}^{l-1} \mathbb{E}\left[\left\langle \nabla F(\theta_{t+i}^{(1)}) - \nabla F(\theta_0), \nabla F(\theta_{t+j}^{(2)}) - \nabla F(\theta_0) \right\rangle \cdot \left\langle \nabla F(\theta_0), \epsilon(\theta_{t+k}^{(2)}) \right\rangle \right]
$$

$$
+ \sum_{i,j,h,k=0}^{l-1} \mathbb{E}\left[\left\langle \nabla F(\theta_{t+i}^{(1)}) - \nabla F(\theta_0), \nabla F(\theta_{t+j}^{(2)}) - \nabla F(\theta_0) \right\rangle \times\right.
$$
$$
\left. \times \left\langle \nabla F(\theta_{t+h}^{(1)}) - \nabla F(\theta_0), \epsilon(\theta_{t+k}^{(2)}) \right\rangle \right]
$$

$$
\leq l^2 ||\nabla F(\theta_0)||^2 \sum_{j,k=0}^{l-1} \mathbb{E}\left[||\nabla F(\theta_{t+j}^{(2)}) - \nabla F(\theta_0)|| \cdot ||\epsilon(\theta_{t+k}^{(2)})|| \right]
$$

$$
+ l ||\nabla F(\theta_0)|| \sum_{j,h,k=0}^{l-1} \mathbb{E}\left[||\nabla F(\theta_{t+j}^{(2)}) - \nabla F(\theta_0)|| \cdot ||\nabla F(\theta_{t+h}^{(1)}) - \nabla F(\theta_0)|| \cdot ||\epsilon(\theta_{t+k}^{(2)})|| \right]
$$

$$
+ l ||\nabla F(\theta_0)|| \sum_{i,j,k=0}^{l-1} \mathbb{E}\left[||\nabla F(\theta_{t+i}^{(1)}) - \nabla F(\theta_0)|| \cdot ||\nabla F(\theta_{t+j}^{(2)}) - \nabla F(\theta_0)|| \cdot ||\epsilon(\theta_{t+k}^{(2)})|| \right]
$$

$$
+ \sum_{i,j,h,k=0}^{l-1} \mathbb{E}\left[||\nabla F(\theta_{t+i}^{(1)}) - \nabla F(\theta_0)|| \cdot ||\nabla F(\theta_{t+j}^{(2)}) - \nabla F(\theta_0)|| \times\right.
$$
$$
\left. \times ||\nabla F(\theta_{t+h}^{(1)}) - \nabla F(\theta_0)|| \cdot ||\epsilon(\theta_{t+k}^{(2)})|| \right]
$$

We bound the four pieces separately. For the first, we can just apply Cauchy-Schwarz and $L$-smoothness, together with Remark 7

$$
\mathbb{E}\left[||\nabla F(\theta_{t+j}^{(2)}) - \nabla F(\theta_0)|| \cdot ||\epsilon(\theta_{t+k}^{(2)})|| \right] \leq L\sqrt{\mathbb{E}\left[||\theta_{t+j}^{(2)} - \theta_0||^2\right] \cdot \mathbb{E}\left[||\epsilon(\theta_{t+k}^{(2)})||^2\right]}
$$
$$
\leq \sqrt{d\sigma_{max}} \cdot L\eta G(t+l)
$$

The bound for the second and third term is equal. We use the conditional independence of the two threads and Lemma 8.

$$
\begin{aligned}
\mathbb{E}\left[||\nabla F(\theta_{t+j}^{(2)}) - \nabla F(\theta_0)|| \cdot ||\nabla F(\theta_{t+h}^{(1)}) - \nabla F(\theta_0)|| \cdot ||\epsilon(\theta_{t+k}^{(2)})||\right] = \\
= \mathbb{E}\left[\mathbb{E}\left[||\nabla F(\theta_{t+j}^{(2)}) - \nabla F(\theta_0)|| \cdot ||\epsilon(\theta_{t+k}^{(2)})|| \mid \mathcal{F}_t\right] \times \right.\\
\left. \times \mathbb{E}\left[||\nabla F(\theta_{t+h}^{(1)}) - \nabla F(\theta_0)|| \mid \mathcal{F}_t\right]\right] \\
\leq \mathbb{E}\left[\sqrt{\mathbb{E}\left[||\nabla F(\theta_{t+j}^{(2)}) - \nabla F(\theta_0)||^2 \mid \mathcal{F}_t\right] \cdot \mathbb{E}\left[||\epsilon(\theta_{t+k}^{(2)})||^2 \mid \mathcal{F}_t\right]} \times \right.\\
\left. \times \mathbb{E}\left[||\nabla F(\theta_{t+h}^{(1)}) - \nabla F(\theta_0)|| \mid \mathcal{F}_t\right]\right] \\
\leq \sqrt{d\sigma_{max}} \cdot \mathbb{E}\left[(L||\theta_t - \theta_0|| + L\eta Gl)^2\right] \\
\leq \sqrt{d\sigma_{max}} \cdot L^2\eta^2 G^2(t+l)^2
\end{aligned}
$$

The last term again makes use of conditional independence and Lemma 8.

$$
\begin{aligned}
\mathbb{E}\left[||\nabla F(\theta_{t+i}^{(1)}) - \nabla F(\theta_0)|| \cdot ||\nabla F(\theta_{t+j}^{(2)}) - \nabla F(\theta_0)|| \right.\\
\left. \times ||\nabla F(\theta_{t+h}^{(1)}) - \nabla F(\theta_0)|| \cdot ||\epsilon(\theta_{t+k}^{(2)})||\right] = \\
= \mathbb{E}\left[\mathbb{E}\left[||\nabla F(\theta_{t+i}^{(1)}) - \nabla F(\theta_0)|| \cdot ||\nabla F(\theta_{t+h}^{(1)}) - \nabla F(\theta_0)|| \mid \mathcal{F}_t\right] \times \right.\\
\left. \times \mathbb{E}\left[||\nabla F(\theta_{t+j}^{(2)}) - \nabla F(\theta_0)|| \cdot ||\epsilon(\theta_{t+k}^{(2)})|| \mid \mathcal{F}_t\right]\right] \\
\leq \mathbb{E}\left[\sqrt{\mathbb{E}\left[||\nabla F(\theta_{t+i}^{(1)}) - \nabla F(\theta_0)||^2 \mid \mathcal{F}_t\right] \cdot \mathbb{E}\left[||\nabla F(\theta_{t+h}^{(1)}) - \nabla F(\theta_0)||^2 \mid \mathcal{F}_t\right]} \times \right.\\
\left. \times \sqrt{\mathbb{E}\left[||\nabla F(\theta_{t+j}^{(2)}) - \nabla F(\theta_0)||^2 \mid \mathcal{F}_t\right] \cdot \mathbb{E}\left[||\epsilon(\theta_{t+k}^{(2)})||^2 \mid \mathcal{F}_t\right]}\right] \\
\leq \sqrt{d\sigma_{max}} \cdot \mathbb{E}\left[(L||\theta_t - \theta_0|| + L\eta Gl)^3\right] \\
\leq \sqrt{d\sigma_{max}} \cdot L^3\eta^3 G^3(t+l)^3
\end{aligned}
$$

The last inequality follows from the use of Remark 7 to bound the moments of $||\theta_t - \theta_0||$ up to order three.

- The upper bound for $VII$ and $VIII$ is the same, even if the error terms are in different positions. Again we invoke conditional independence to get rid of the dot products that only contain $\nabla F(\theta_0)$,

and subsequently apply Cauchy-Schwarz inequality.

$$\mathbb{E}\left[\left\langle \sum_{i=0}^{l-1}\nabla F(\theta_{t+i}^{(1)}), \sum_{j=0}^{l-1}\nabla F(\theta_{t+j}^{(2)})\right\rangle \cdot \left\langle \sum_{i=0}^{l-1}\epsilon(\theta_{t+i}^{(1)}), \sum_{j=0}^{l-1}\epsilon(\theta_{t+j}^{(2)})\right\rangle\right]$$

$$= \sum_{i,j,h,k=0}^{l-1}\mathbb{E}\left[\left\langle \nabla F(\theta_0) + \left(\nabla F(\theta_{t+i}^{(1)}) - \nabla F(\theta_0)\right),\right.\right.$$

$$\left.\left.\nabla F(\theta_0) + \left(\nabla F(\theta_{t+j}^{(2)}) - \nabla F(\theta_0)\right)\right\rangle \cdot \left\langle \epsilon(\theta_{t+h}^{(1)}), \epsilon(\theta_{t+k}^{(2)})\right\rangle\right]$$

$$= \sum_{i,j,h,k=0}^{l-1}\mathbb{E}\left[\left\langle \nabla F(\theta_{t+i}^{(1)}) - \nabla F(\theta_0), \nabla F(\theta_{t+j}^{(2)}) - \nabla F(\theta_0)\right\rangle \cdot \left\langle \epsilon(\theta_{t+h}^{(1)}), \epsilon(\theta_{t+k}^{(2)})\right\rangle\right]$$

$$\leq L^2 \sum_{i,j,h,k=0}^{l-1}\mathbb{E}\left[||\theta_{t+i}^{(1)} - \theta_0|| \cdot ||\theta_{t+j}^{(2)} - \theta_0|| \cdot ||\epsilon(\theta_{t+h}^{(1)})|| \cdot ||\epsilon(\theta_{t+k}^{(2)})||\right]$$

$$\leq L^2 \sum_{i,j,h,k=0}^{l-1}\mathbb{E}\left[\mathbb{E}\left[||\theta_{t+i}^{(1)} - \theta_0|| \cdot ||\epsilon(\theta_{t+h}^{(1)})|| \mid \mathcal{F}_t\right] \times\right.$$

$$\left.\times \mathbb{E}\left[||\theta_{t+j}^{(2)} - \theta_0|| \cdot ||\epsilon(\theta_{t+k}^{(2)})|| \mid \mathcal{F}_t\right]\right]$$

$$\leq L^2 \sum_{i,j,h,k=0}^{l-1}\mathbb{E}\left[\sqrt{\mathbb{E}\left[||\theta_{t+i}^{(1)} - \theta_0||^2 \mid \mathcal{F}_t\right] \cdot \mathbb{E}\left[||\epsilon(\theta_{t+h}^{(1)})||^2 \mid \mathcal{F}_t\right]} \times\right.$$

$$\left.\times \sqrt{\mathbb{E}\left[||\theta_{t+j}^{(2)} - \theta_0||^2 \mid \mathcal{F}_t\right] \cdot \mathbb{E}\left[||\epsilon(\theta_{t+k}^{(2)})||^2 \mid \mathcal{F}_t\right]}\right]$$

$$\leq l^4 L^2 \eta^2 G^2 (t+l)^2 d\sigma_{max}$$

- Also the upper bounds for $IX$ and $X$ are equal. In the first one, when $k \neq j$ we can condition on $\mathcal{F}_{t+l}^{(1)}$ and $\mathcal{F}_{t+\max\{k,j\}}^{(2)}$ to get that the expectation is null. Then we are only left with a sum on three indexes $i, j, h$ and $k = j$. In the last passage we again condition on the appropriate $\sigma$-algebras to bound separately the two threads.

$$\mathbb{E}\left[\left\langle \sum_{i=0}^{l-1}\nabla F(\theta_{t+i}^{(1)}), \sum_{j=0}^{l-1}\epsilon(\theta_{t+j}^{(2)})\right\rangle \cdot \left\langle \sum_{h=0}^{l-1}\epsilon(\theta_{t+h}^{(1)}), \sum_{k=0}^{l-1}\epsilon(\theta_{t+k}^{(2)})\right\rangle\right]$$

$$= \sum_{i,j,h=0}^{l-1}\mathbb{E}\left[\left\langle \nabla F(\theta_0) + \left(\nabla F(\theta_{t+i}^{(1)}) - \nabla F(\theta_0)\right), \epsilon(\theta_{t+j}^{(2)})\right\rangle \cdot \left\langle \epsilon(\theta_{t+h}^{(1)}), \epsilon(\theta_{t+j}^{(2)})\right\rangle\right]$$

$$= \sum_{i,j,h=0}^{l-1}\mathbb{E}\left[\left\langle \nabla F(\theta_{t+i}^{(1)}) - \nabla F(\theta_0), \epsilon(\theta_{t+j}^{(2)})\right\rangle \cdot \left\langle \epsilon(\theta_{t+h}^{(1)}), \epsilon(\theta_{t+j}^{(2)})\right\rangle\right]$$

$$\leq \sum_{i,j,h=0}^{l-1}\mathbb{E}\left[||\nabla F(\theta_{t+i}^{(1)}) - \nabla F(\theta_0)|| \cdot ||\epsilon(\theta_{t+j}^{(2)})||^2 \cdot ||\epsilon(\theta_{t+h}^{(1)})||\right]$$

$$\leq \sum_{i,j,h=0}^{l-1}\mathbb{E}\left[\mathbb{E}\left[||\nabla F(\theta_{t+i}^{(1)}) - \nabla F(\theta_0)|| \cdot ||\epsilon(\theta_{t+h}^{(1)})|| \mid \mathcal{F}_t\right] \cdot \mathbb{E}\left[||\epsilon(\theta_{t+j}^{(2)})||^2 \mid \mathcal{F}_t\right]\right]$$

$$\leq l^3 L\eta G(t+l) (d\sigma_{max})^{3/2}$$

We put together all these upper bounds, leaving in extended form all the terms that are more significant than $O(\eta^2(t+l)^2)$. We get

$$
\begin{aligned}
Var\,(Q_1) &= \mathbb{E}\left[Q_1^2\right] - \mathbb{E}\left[Q_1\right]^2 \\
&\lesssim \frac{2||\nabla F(\theta_0)||^2 d\sigma_{max}}{l} + \frac{d^2\sigma_{max}^2}{l^2} \\
&\quad + \eta \cdot \left(\frac{4d\sigma_{max}||\nabla F(\theta_0)||LG(t+l)}{l} + \frac{2LG(t+l)(d\sigma_{max})^{3/2}}{l}\right) \\
&\quad + \eta \cdot \left(8LG||\nabla F(\theta_0)||^3(t+l) + 2||\nabla F(\theta_0)||^2LG(t+l)\sqrt{d\sigma_{max}}\right) + O(\eta^2(t+l)^2)
\end{aligned}
$$

which immediately translates to a bound for the standard deviation of the following form

$$
\begin{aligned}
\text{sd}\,(Q_1) &\lesssim \frac{||\nabla F(\theta_0)||\sqrt{2d\sigma_{max}}}{\sqrt{l}} + \frac{d\sigma_{max}}{l} \\
&\quad + \sqrt{\eta} \cdot \left(8LG||\nabla F(\theta_0)||^3(t+l) + 2||\nabla F(\theta_0)||^2LG(t+l)\sqrt{d\sigma_{max}}\right)^{1/2} \\
&\quad + \sqrt{\eta} \cdot \left(\frac{4d\sigma_{max}||\nabla F(\theta_0)||LG(t+l)}{l} + \frac{2LG(t+l)(d\sigma_{max})^{3/2}}{l}\right)^{1/2} + O(\eta(t+l))
\end{aligned}
\tag{19}
$$

We combine (19) with the fact, consequence of (16), that $\mathbb{E}[Q_1]/||\nabla F(\theta_0)||^2 \gtrsim 1 + O(\eta(t+l))$, to get the desired inequality

$$
\text{sd}(Q_1) \lesssim C_1(\eta,l) \cdot \mathbb{E}[Q_1]
$$

where

$$
\begin{aligned}
C_1(\eta,l) &= \frac{1}{||\nabla F(\theta_0)||^2} \cdot \left\{ \frac{||\nabla F(\theta_0)||\sqrt{2d\sigma_{max}}}{\sqrt{l}} + \frac{d\sigma_{max}}{l} \right. \\
&\quad + \sqrt{\eta} \cdot \left(8LG||\nabla F(\theta_0)||^3(t+l) + 2||\nabla F(\theta_0)||^2LG(t+l)\sqrt{d\sigma_{max}}\right)^{1/2} \\
&\quad \left. + \sqrt{\eta} \cdot \left(\frac{4d\sigma_{max}||\nabla F(\theta_0)||LG(t+l)}{l} + \frac{2LG(t+l)(d\sigma_{max})^{3/2}}{l}\right)^{1/2} \right\}
\end{aligned}
$$

This confirms that $C_1(\eta,l) = O(1/\sqrt{l}) + O(\sqrt{\eta(t+l)})$.

## G   Proof of Theorem 3

As before, we only consider $Q_1$ for simplicity. To provide an upper bound for $|\mathbb{E}[Q_1]|$, we use the fact that $\nabla F(\theta^*) = 0$ together with Assumption 3.2. Starting from (15) we have

$$
\begin{aligned}
|\mathbb{E}\left[Q_1\right]| &= \frac{1}{l^2} \left| \sum_{j=0}^{l-1}\sum_{k=0}^{l-1} \mathbb{E}\left[ \langle \nabla F(\theta_{t+j}^{(1)}), \nabla F(\theta_{t+k}^{(2)}) \rangle \right] \right| \\
&\leq \frac{1}{l^2} \sum_{j=0}^{l-1}\sum_{k=0}^{l-1} \mathbb{E}\left[ ||\nabla F(\theta_{t+j}^{(1)}) - \nabla F(\theta^*)|| \cdot ||\nabla F(\theta_{t+k}^{(2)}) - \nabla F(\theta^*)|| \right] \\
&\leq \frac{L^2}{l^2} \sum_{j=0}^{l-1}\sum_{k=0}^{l-1} \mathbb{E}\left[ ||\theta_{t+j}^{(1)} - \theta^*|| \cdot ||\theta_{t+k}^{(2)} - \theta^*|| \right] \\
&\leq \frac{L^2}{l^2} \sum_{j=0}^{l-1}\sum_{k=0}^{l-1} \sqrt{ \mathbb{E}\left[ ||\theta_{t+j}^{(1)} - \theta^*||^2 \right] \cdot \mathbb{E}\left[ ||\theta_{t+k}^{(2)} - \theta^*||^2 \right] }
\end{aligned}
$$

Now we can use Lemma 4 that states that, for $\eta \leq \frac{\mu}{L^2}$,

$$\mathbb{E}\left[||\theta_t - \theta^*||^2\right] \leq \left(1 - 2\eta(\mu - L^2\eta)\right)^t \cdot \mathbb{E}\left[||\theta_0 - \theta^*||^2\right] + \frac{G^2\eta}{\mu - L^2\eta}. \tag{20}$$

As $t \to \infty$ we have that $\mathbb{E}\left[||\theta_t - \theta^*||^2\right] \lesssim \frac{G^2\eta}{\mu - L^2\eta}$. $L$-smoothness combined with (20) also gets

$$\mathbb{E}\left[||\nabla F(\theta_t)||^2\right] \lesssim \frac{L^2 G^2 \eta}{\mu - L^2\eta} \qquad \text{as} \ \ t \to \infty. \tag{21}$$

Since the first term of (20) is decreasing in $t$, our bound on the expectation of $Q_1$ is

$$|\mathbb{E}\left[Q_1\right]| \leq L^2 \cdot \left(\left(1 - 2\eta(\mu - L^2\eta)\right)^t \cdot \mathbb{E}\left[||\theta_0 - \theta^*||^2\right] + \frac{G^2\eta}{\mu - L^2\eta}\right) \tag{22}$$

To deal with the second moment, we introduce the notation

$$S_k := \sum_{i=0}^{l-1} g(\theta_{t+i}^{(k)}, Z_{t+i+1}^{(k)}) = \sum_{i=0}^{l-1} \nabla F(\theta_{t+i}^{(k)}) + \sum_{i=0}^{l-1} \epsilon(\theta_{t+i}^{(k)}) =: G_k + e_k.$$

where $G_k$ is the true signal in the first window of thread $k$ and $e_k$ the related noise. Conditional on $\mathcal{F}_t$, the random variables $S_1$ and $S_2$ are independent and identically distributed. Then we can write

$$\begin{aligned}
l^4 \cdot \mathbb{E}[Q_1^2] = \mathbb{E}\left[\langle S_1, S_2 \rangle^2\right] &= \mathbb{E}\left[S_2^T S_1 S_1^T S_2\right] \\
&= \mathbb{E}\left[\text{Tr}(S_2^T S_1 S_1^T S_2)\right] = \mathbb{E}\left[\text{Tr}(S_1 S_1^T S_2 S_2^T)\right] \\
&= \text{Tr}\left(\mathbb{E}\left[S_1 S_1^T S_2 S_2^T\right]\right) = \text{Tr}\left(\mathbb{E}\{\mathbb{E}\left[S_1 S_1^T \mid \mathcal{F}_t\right] \cdot \mathbb{E}\left[S_2 S_2^T \mid \mathcal{F}_t\right]\}\right) \\
&= \text{Tr}\left(\mathbb{E}\{\mathbb{E}\left[S_1 S_1^T \mid \mathcal{F}_t\right]^2\}\right)
\end{aligned}$$

The goal is now to show that the matrix $\mathbb{E}\left[S_1 S_1^T \mid \mathcal{F}_t\right]$ is positive definite, and provide a lower bound for its second moment using the fact that if $A \succeq \lambda I$ for $\lambda \geq 0$, then $A^2 \succeq \lambda^2 I$. We can write

$$\begin{aligned}
\mathbb{E}\left[S_1 S_1^T \mid \mathcal{F}_t\right] &= \mathbb{E}\left[(G_1 + e_1)(G_1 + e_1)^T \mid \mathcal{F}_t\right] \\
&= \mathbb{E}\left[G_1 G_1^T \mid \mathcal{F}_t\right] + \mathbb{E}\left[G_1 e_1^T \mid \mathcal{F}_t\right] + \mathbb{E}\left[e_1 G_1^T \mid \mathcal{F}_t\right] + \mathbb{E}\left[e_1 e_1^T \mid \mathcal{F}_t\right]
\end{aligned}$$

We immediately have that $\mathbb{E}\left[G_1 G_1^T \mid \mathcal{F}_t\right] \succeq 0$, because, for any $x \in \mathbb{R}^d$,

$$x^T \mathbb{E}\left[G_1 G_1^T \mid \mathcal{F}_t\right] x = \mathbb{E}\left[x^T G_1 G_1^T x \mid \mathcal{F}_t\right] = \mathbb{E}\left[||x^T G_1||^2 \mid \mathcal{F}_t\right] \geq 0.$$

Moreover we can also find an easy lower bound for the error term using Assumption 3.3,

$$\begin{aligned}
\mathbb{E}\left[e_1 e_1^T \mid \mathcal{F}_t\right] &= \mathbb{E}\left[\left(\sum_{i=0}^{l-1} \epsilon(\theta_{t+i}^{(1)})\right)\left(\sum_{j=0}^{l-1} \epsilon(\theta_{t+j}^{(1)})\right)^T \Bigg| \mathcal{F}_t\right] \\
&= \sum_{i=0}^{l-1} \mathbb{E}\left\{\mathbb{E}\left[\epsilon(\theta_{t+i}^{(1)})\epsilon(\theta_{t+i}^{(1)})^T \mid \mathcal{F}_{t+i-1}\right]\right\} \\
&\succeq l \cdot \sigma_{min} \cdot I
\end{aligned}$$

To lower bound the remaining terms we introduce a simple Lemma.

**Lemma 12** *If $u, v \in \mathbb{R}^d$, then $uv^T + vu^T \succeq -2||u|| \cdot ||v|| \cdot I$*

**Proof** We apply the Cauchy-Schwarz inequality and get, for any $x \in \mathbb{R}^d$,

$$x^T(uv^T + vu^T + 2||u|| \cdot ||v|| \cdot I)x = x^T uv^T x + x^T vu^T x + 2||u|| \cdot ||v|| \cdot x^T x$$
$$= \langle x, u \rangle \langle v, x \rangle + \langle x, v \rangle \langle u, x \rangle + 2||u|| \cdot ||v|| \cdot ||x||^2 \geq 0$$

∎

Using Lemma 12, and Lemma 10 *ii)* in the last inequality, we immediately get that

$$\mathbb{E}\left[G_1 e_1^T \ |\mathcal{F}_t\right] + \mathbb{E}\left[e_1 G_1^T \ |\mathcal{F}_t\right] \succeq -2\mathbb{E}\left[||G_1|| \cdot ||e_1|| \ |\mathcal{F}_t\right] \cdot I$$

$$\succeq -2\sum_{i=1}^{l}\sum_{j=1}^{l} \mathbb{E}\left[||\nabla F(\theta_{t+i}^{(1)})|| \cdot ||\epsilon(\theta_{t+j}^{(1)})|| \ |\mathcal{F}_t\right] \cdot I$$

$$\succeq -2\sum_{i=0}^{l-1}\sum_{j=0}^{l-1} \sqrt{\mathbb{E}\left[||\nabla F(\theta_{t+i}^{(1)})||^2 \ |\mathcal{F}_t\right] \cdot \mathbb{E}\left[||\epsilon(\theta_{t+j}^{(1)})||^2 \ |\mathcal{F}_t\right]} \cdot I$$

$$\succeq -2l^2 \cdot \sqrt{d\sigma_{max}} \cdot (L||\theta_t - \theta^*|| + L\eta Gl) \cdot I$$

Notice that we could improve the bound using the fact that $\epsilon(\theta_{t+j}^{(1)})$ is independent from $\nabla F(\theta_{t+i}^{(1)})$ for any $j \geq i$. Putting the pieces together we get that

$$\mathbb{E}\left[S_1 S_1^T \ |\mathcal{F}_t\right] \succeq \left(l\sigma_{min} - 2l^2 \cdot \sqrt{d\sigma_{max}} \cdot (L||\theta_t - \theta^*|| + L\eta Gl)\right) \cdot I$$

$$\Rightarrow \quad \mathbb{E}\left[S_1 S_1^T \ |\mathcal{F}_t\right]^2 \succeq \left(l\sigma_{min} - 2l^2 \cdot \sqrt{d\sigma_{max}} \cdot (L||\theta_t - \theta^*|| + L\eta Gl)\right)^2 \cdot I$$

$$\succeq \left\{ l^2\sigma_{min}^2 + 4l^4 d\sigma_{max} \cdot (L||\theta_t - \theta^*|| + L\eta Gl)^2 \right.$$

$$\left. - 4l^3\sigma_{min}\sqrt{d\sigma_{max}} \cdot (L||\theta_t - \theta^*|| + L\eta Gl) \right\} \cdot I$$

and then, using the asymptotic bound in (20),

$$\mathbb{E}\left[\mathbb{E}\left[S_1 S_1^T \ |\mathcal{F}_t\right]^2\right] \succeq \left\{ l^2\sigma_{min}^2 - 4l^3\sigma_{min}\sqrt{d\sigma_{max}} \cdot (L \cdot \mathbb{E}\left[||\theta_t - \theta^*||\right] + L\eta Gl) \right\} \cdot I$$

$$\overset{t \to \infty}{\succeq} \left\{ l^2\sigma_{min}^2 - 4l^3\sigma_{min}\sqrt{d\sigma_{max}} \cdot \left(\frac{LG\sqrt{\eta}}{\sqrt{\mu - L^2\eta}} + L\eta Gl\right) \right\} \cdot I$$

which finally gives the bound on the second moment, which is

$$l^4 \cdot \mathbb{E}[Q_1^2] \gtrsim d \cdot \left(l^2\sigma_{min}^2 - 4l^3\sigma_{min}\sqrt{d\sigma_{max}}LG\sqrt{\eta} \cdot \left(\frac{1}{\sqrt{\mu - L^2\eta}} + l\sqrt{\eta}\right)\right)$$

$$\geq dl^2\sigma_{min}^2 - K_1 l^3\sqrt{\eta} - K_2 l^4\eta$$

Using the fact shown before, that

$$\mathbb{E}[Q_1]^2 \lesssim \frac{L^4 G^4 \eta^2}{(\mu - L^2\eta)^2} \qquad \text{as} \ \ t \to \infty,$$

we can bound the variance of $Q_1$ from below with

$$Var(Q_1) = \mathbb{E}[Q_1^2] - \mathbb{E}[Q_1]^2 \geq \frac{d\sigma_{min}^2}{l^2} - \frac{K_1\sqrt{\eta}}{l} - K_2\eta - \frac{L^4 G^4 \eta^2}{(\mu - L^2\eta)^2}$$

and then

$$Var(Q_1) \gtrsim \left(\frac{d\sigma_{min}^2}{l^2} - \frac{K_1\sqrt{\eta}}{l} + O(\eta)\right) \cdot \frac{\mathbb{E}[Q_1]^2(\mu - L^2\eta)^2}{L^4 G^4 \eta^2}.$$

The desired inequality is finally

$$|\mathbb{E}[Q_1]| \lesssim C_2(\eta) \cdot \mathrm{sd}(Q_1)$$

with

$$C_2(\eta) = \frac{L^2 G^2 \eta}{(\mu - L^2 \eta)} \cdot \left( \frac{d\sigma_{min}^2}{l^2} - \frac{K_1 \sqrt{\eta}}{l} + O(\eta) \right)^{-1/2} = C_2 \cdot \eta + o(\eta).$$

