# OpenReview forum: "Robust Learning Rate Selection for Stochastic Optimization via Splitting Diagnostic"
_TMLR — Accepted by TMLR_

### Review · Reviewer_dxDg · 2023-07-17

**Summary Of Contributions:**

This work proposes a learning rate selection method with the suggested gradient coherence criterion

**Audience:**

Yes

**Broader Impact Concerns:**

It is fine

**Claims And Evidence:**

Yes

**Requested Changes:**

I would like to see the same experiment in Figure 6 on ImageNet for resnet

**Strengths And Weaknesses:**

The method is demonstrated to work well, along with some theory that analyzes the theory for convex problems. Since the main motivation is about applying the method to deep learning, I personally do not think the theory aspect is relevant or convincing. Theoretical works suggested that the dynamics of SGD is predominantly saddle-to-saddle, whereas there is no saddle point in a convex landscape. However, I do not think there is any problem as long as the method is demonstrated to work well

See the requested changes section

---

> ### Author Response · Authors · 2023-11-21
> **Addressing the reviewer's comments**
>
> * A new experiment on a more complex dataset
>
> Training various networks on Imagenet requires a non-trivial amount of computing infrastructure and run time, which unfortunately is not available to us. As we believe the reviewer was looking for results for a more complex classification task, we have ran several simulations on Cifar100, a vision classification tasks with 100 classes (rather than 1000 as on Imagenet). The results of these experiments are shown in the table below, where for each method and each initial learning rate we report the maximum test accuracy achieved before a certain epoch. Two things are clear from this experiment:
>
> 1. SplitSGD is able to get to a higher test accuracy than SGD with fixed learning rate and Adam in this setting.
> 2. Independently from the starting learning rate (at least for values chosen in a reasonable range) the performances of SplitSGD are extremely consistent, due to the adaptive nature of its learning rate selection procedure.
>
> | *method* | *lr* | *max_100* | *max_150* | *max_200* | *max_250* | *max_300* | *max_350* |
> | ---------- | ------ | ----------- | ----------- | ----------- | ----------- | ----------- | ----------- |
> | Adam       | 0.005  | 58.33       | 58.81       | 58.81       | 59.51       | 59.57       | 59.57       |
> | SGD        | 0.005  | 68.86       | 68.99       | 68.99       | 68.99       | 69.33       | 70.45       |
> | SplitSGD   | 0.005  | 70.63       | 71.04       | 71.04       | 72.35       | 73.0        | 73.16       |
> | Adam       | 0.01   | 60.1        | 60.33       | 61.01       | 61.01       | 61.01       | 61.02       |
> | SGD        | 0.01   | 67.74       | 67.74       | 67.74       | 67.74       | 67.74       | 67.74       |
> | SplitSGD   | 0.01   | 70.66       | 70.82       | 73.18       | 73.51       | 73.6        | 73.78       |
> | Adam       | 0.03   | 61.79       | 61.79       | 62.29       | 62.59       | 62.88       | 62.88       |
> | SGD        | 0.03   | 65.08       | 65.84       | 65.84       | 65.94       | 65.94       | 65.94       |
> | SplitSGD   | 0.03   | 69.58       | 72.06       | 72.06       | 72.06       | 72.67       | 72.81       |
> | Adam       | 0.1    | 53.08       | 53.32       | 53.38       | 53.85       | 53.85       | 54.3        |
> | SGD        | 0.1    | 62.04       | 62.53       | 62.53       | 62.53       | 62.53       | 62.53       |
> | SplitSGD   | 0.1    | 69.88       | 69.88       | 70.11       | 70.15       | 70.15       | 72.49       |

---

### Review · Reviewer_ocBo · 2023-07-24

**Summary Of Contributions:**

The paper proposes a heuristic to detect proximity to stationarity in a stochastic optimization setting. The main idea is to start two simultaneous threads of SGD and measure the correlation between the directions of the gradients, where no correlation indicates proximity to stationarity. While measuring the correlation was done in previous works, the main novelty in this paper stems from the method in which stationarity is detected, which evaluates the percentage of negative versus positive correlations. Experimentation is performed to test the performance of the proposed algorithm SplitSGD compared to other commonly used optimization algorithms.

**Audience:**

Yes

**Broader Impact Concerns:**

None.

**Claims And Evidence:**

No

**Requested Changes:**

- In a few figures in the paper (e.g. Figure 1), much of the text explaining the figures appears in the body of the text but not in the caption of the figures which I find to be inconvenient.

- The nature of Assumptions 3.3 and 3.4 isn't clear. When should we expect these assumptions to hold? Why are they not extremely limiting?

- While the experiments highlight certain cases in which SplitSGD is preferable over the compared methods, it is not obvious that this is indeed an indication of a broader phenomenon. For example, this can just be an artifact of fine-tuning the hyperparameters of SplitSGD to the point where they outperform other methods, whereas it is possible that a similar refinement for other methods could yield better performance. Specifically, I wasn't convinced that the experimental part on neural networks provides better performance overall, and I think it would be better to first experiment and study this algorithm in a simpler setting (e.g. a setting which mildly relaxes the theoretical assumptions in Section 3).

- The summary states that "We have developed an efficient optimization method called SplitSGD" -- I wouldn't argue that this optimization method is efficient based solely on the experiments performed in this paper and I would suggest to omit the word "efficient".

**Strengths And Weaknesses:**

Strengths:

- The paper is written in a clear way.

- The paper seems to come up with an interesting way of encapsulating the somewhat complex step size selection schedule problem into a single parameter $q$. This potentially simplifies the hyperparameter selection process without making significant compromises in its optimization efficacy.


Weaknesses:

- While I appreciate the theoretical work done in the paper, it only seems to provide an initial understanding of SplitSGD (namely, proof of its asymptotic convergence) under strong assumptions. In light of this missing theoretical understanding, I believe the main criteria to judge the paper by is its experimental parts, however I also found this part of the paper to be somewhat lacking:

- Unfortunately, I didn't find the experimentation in the paper convincing enough to merit acceptance. It is difficult to assess the practical performance of a new algorithm based on the experiments performed in the scope of a single paper in such an instance. See requested changes section below for further detail.

- While the parameter $q$ encapsulates the step size selection schedule in a neat way, the algorithm does introduce other new hyperparameters that might require some amount of tweaking (e.g. $w$ and $l$). It is therefore not clear if this indeed simplifies the optimization process beyond that of other algorithms that require finetuning the step size schedule.

---

> ### Author Response · Authors · 2023-11-21
> **Addressing the reviewer's comments**
>
> * Good SplitSGD performances due to hyper-parameters fine tuning.
>
> After finding an acceptable range for the parameters q and w in the two broad settings considered (regressions and Deep Neural Network) we simply applied the same set of parameters in all experiments, and shown in Figure 8 that some deviation from those parameters does not really impact the performance of SplitSGD by much.
>
> We have also ran another version of Figure 8 using ResNet on Cifar 10, and the results are consistent with what we had already observed for the FNN and CNN, meaning that a pretty large variation in the choice of the q and w parameters (q $\in$ {0.15, 0.25, 0.35} and w $\in$ {2, 4, 8}) does not change the performance of SplitSGD significantly. Below we show some results for the new comparison, where for each pair (q, w) we ran 4 simulations of 350 epochs each, and at several different checkpoints (epoch 100, 150, 200, 250 and 300) we looked at the maximum test accuracy obtained before. The numbers reported are the average and standard deviations of these estimates. We can appreciate the fact that all sets of parameters gave very similar results.
>
> | *q* | *w* | *max_100-mean* | *max_100-std* | *max_150-mean* | *max_150-std* | *max_200-mean* | *max_200-std* | *max_250-mean* | *max_250-std* | *max_300-mean* | *max_300-std* | *max_350-mean* | *max_350-std* |
> | ----- | ----- | ---------------- | --------------- | ---------------- | --------------- | ---------------- | --------------- | ---------------- | --------------- | ---------------- | --------------- | ---------------- | --------------- |
> | 0.15  | 4     | 92.4             | 0.52            | 92.55            | 0.62            | 92.64            | 0.6             | 92.65            | 0.59            | 92.65            | 0.59            | 92.66            | 0.6             |
> | 0.25  | 2     | 92.38            | 0.22            | 92.62            | 0.15            | 93.03            | 0.31            | 93.1             | 0.25            | 93.22            | 0.21            | 93.31            | 0.22            |
> | 0.25  | 8     | 92.64            | 0.46            | 92.8             | 0.42            | 92.85            | 0.41            | 92.88            | 0.45            | 92.88            | 0.46            | 92.88            | 0.46            |
> | 0.35  | 4     | 92.49            | 0.29            | 92.85            | 0.18            | 92.9             | 0.17            | 93.07            | 0.05            | 93.3             | 0.16            | 93.3             | 0.16            |
>
> * The limiting nature of Assumptions 3.3 and 3.4.
>
> We think that assumptions 3.3 and 3.4 are not unreasonable and are actually pretty standard. For example, assumption H7 in [2] corresponds to our Assumption 3.4. Moreover, the upper bound for 3.3 is a standard assumption (the eigenvalue of a rank-one matrix is finite). It is possible that the lower bound in Assumption 3.4 is not needed, although this would change the proof of Theorem 3 to a degenerate case.
>
> [2] Non-asymptotic analysis of stochastic approximation algorithms for machine learning, Moulines, Eric and Bach, Francis, Advances in neural information processing systems, 2011
>
> * Minor changes.
>
> We will update the figures' captions and body of the text according to the suggestions.

---

### Review · Reviewer_5Pef · 2023-11-10

**Summary Of Contributions:**

**Summary:** This paper constructs an adaptive schedule for the learning rate that detects when a fixed stepsize SGD hits a stationary distribution and then geometrically decreasing the stepsize. The authors consider this in the setting, $\min_{\theta} F(\theta) = E[ f(\theta, Z)]$ where Z ~ (X,y) is a datapoint and provides theoretical guarantees when $F$ is strongly convex and smooth ($L$-Lipschitz gradients) with $f$ convex. In addition, it is assumed that the gradient $\mathbb{E}[ \| g(\theta_t, Z_t+1)\|^4 \, | \, Z_1,..., Z_t] $ is bounded and the error $\epsilon_t = g(\theta_{t-1}, Z_t) - \nabla F(\theta_{t-1})$ between the stochastic gradient, $g(\theta_{t-1}, Z_t)$ and its mean has mean $0$ and covariance bounded above and away from $0$.  \\

The main algorithm called SplitSGD runs two SGD threads initialized at the same iterate using independent data points and then performs a hypothesis test to determine if SGD with fixed stepsize has reached stationarity. The two threads each run $w$ windows of length $\ell$ and compute the average gradient over $\ell$ with the fixed stepsize. Then the similarity between the two averaged gradient threads is computed, $Q_i$ is the inner product between (avg.) $g_i^1$ and $g_i^2$ where $i = 1,..., w$.  The idea being that if one is at stationarity then these two gradients will be nearly $0$ (i.e., orthogonal) and otherwise (not near stationarity) point in the same direction. Essentially, $Q_i$ is trying to approximate $\|\nabla F(\theta_t)\|^2$.To avoid the issue of determining what nearly $0$ means, the authors instead consider the $\text{sign}(Q_i)$:

$$
T_D =  S \text{ if $\sum_{i=1}^w (1-\text{sign}(Q_i)) /2 \ge q \cdot w$ and } N,  \text{otherwise},
$$

where $q \in [0,1]$.
If $T_D = S$ is returned, then one is near stationarity and the window $w$ is increased and stepsize is decreased. On the other hand, if $N$ is returned then all is kept the same. This idea is similar to other works including [Chee and Toulis].

The authors prove that SplitSGD is guaranteed to converge with probability $1$ provided the number of diagnostics is going to $\infty$. They do not have a convergence rate. The authors also provide numerical experiments showing good performance of their algorithm against other similar "stationary" distribution detection algorithms.

The proofs are correct (as far as I could tell). The idea is quite similar to [Chee and Toulis] and so this should be taken into account, but the authors did address the differences namely the windows and "forward" instead of "along the way" computing the gradients. My biggest concern is twofold: 1). This idea, I suppose, should fail for the least squares problem when the number of samples is larger than parameter, i.e., over-parameterized regime (see below) and 2). Missing in the experiments and theory is the actual algorithm being employed, for instance is one using "streaming" (or one pass of the data)?, Mini-batching?

Overall, I believe the paper is good with an interesting idea. The main convergence theorem is not there yet, but it is a sufficient idea. I would stress that the experiments should be made more precise especially given that the main theorem is not proven yet.

**Audience:**

Yes

**Broader Impact Concerns:**

I have no broader impact concerns.

**Claims And Evidence:**

Yes

**Requested Changes:**

**Some (small) comments/suggestions below**
1. The presentation of the paper could be better. It was a little difficult to follow the windows, diagonistic, and how it all fits together to update the iterates.
2. As mentioned above the experiment, should be made more precise. The captions of the Figures do include enough details to follow and the figures themselves do not say enough. For instance, in Figure 5, the y-axis is missing on the top figures. Moreover the caption for them just reads comparison between Splitting and pflug diagonsitic. Not enough explanation for what the reader is suppose to learn from this figure.

**Strengths And Weaknesses:**

**Major comments/concerns**

1. The simple experiments (e.g., logistic and least squares) are all done in the under-parameterized regime (i.e., the number of samples is larger than the dimension). One would expect with Gaussian data that in the over-parameterized regime that a \textit{fixed} stepsize is sufficient for convergence. I don't this algorithm would beat the best fixed stepsize (i.e., the one that converges in this regime). Can the authors comment on this and produce some figure?
2. One should also make sure things are "dimension-independent". For instance, take two iid Gaussian of length $d$, there inner product on average is $1/\sqrt{d}$. Things should be normalized so that as dimension grows you are not just getting the orthogonality effects just because of dimension.
3.  In the experiments, it is not clear exactly what algorithm is used w.r.t. the data (i.e., streaming or multi-pass) and mini-batching? I would expect that as the mini-batch increases the stepsize should not decrease and a fixed stepsize is used. Is this correct?

1. The simple experiments (e.g., logistic and least squares) are all done in the under-parameterized regime (i.e., the number of samples is larger than the dimension). One would expect with Gaussian data that in the over-parameterized regime that a \textit{fixed} stepsize is sufficient for convergence. I don't this algorithm would beat the best fixed stepsize (i.e., the one that converges in this regime). Can the authors comment on this and produce some figure?
2. One should also make sure things are "dimension-independent". For instance, take two iid Gaussian of length $d$, there inner product on average is $1/\sqrt{d}$. Things should be normalized so that as dimension grows you are not just getting the orthogonality effects just because of dimension.
3.  In the experiments, it is not clear exactly what algorithm is used w.r.t. the data (i.e., streaming or multi-pass) and mini-batching? I would expect that as the mini-batch increases the stepsize should not decrease and a fixed stepsize is used. Is this correct?

---

> ### Author Response · Authors · 2023-11-21
> **Addressing the reviewer's comments**
>
> * Performance of SplitSGD against best fixed stepsize in over-parameterized regime.
>
> This is a good point, and we agree that in the over-parameterized settings there might exist a best fixed learning rate. However, even in that setting the problem of finding (or getting close to) this optimal learning rate exists. We argue that our approach with adaptive learning rate selection might not beat the best fixed learning rate, but can be beneficial in getting sufficiently close to it without having to manually tune it.
>
> * Normalization to not get orthogonality just because of dimension when this grows.
>
> We agree with the reviewer that, since the absolute value of the inner product between two vectors in high dimension scales as $\frac{1}{\sqrt{d}}$, any positive or negative threshold for convergence would need to be rescaled accordingly. However, in the Splitting Diagnostic we use the sign of the dot product rather than the magnitude, which would not be impacted by the rescaling. In addition, we empirically observe that in high-dimensional settings (the DNN experiments, Section 4.2) the stationarity detection of SplitSGD does not seem to suffer from high-dimensionality issues.
>
> * Details of the SplitSGD algorithm used for experiments (streaming, mini-batch, etc).
>
> We thank the reviewer for pointing this out and we apologize for the lack of clarity on this aspect in the paper. In the linear and logistic regression experiments, we use the streaming version of SGD where the batch size is simply set to 1, meaning that each data point in the training set is observed individually. In the deep learning experiments this would not be computationally feasible, so we used a default batch size for all the optimization methods on each dataset (which is generally dictated by the GPU memory availability). We believe the reviewer is correct in pointing out that increasing the batch size would result in not needing to decrease the stepsize, similarly to what noted in [1]
> This batch size was not fine tuned but just selected based on what was recommended for each dataset, so we used 128 for ResNet on Cifar10 and Cifar100, 64 for the CNN on Fashion-MNIST, 20 for the LSTM.
>
> [1] "Don't Decay the Learning Rate, Increase the Batch Size", Samuel L. Smith and Pieter-Jan Kindermans and Quoc V. Le, ICLR 2018, \url{https://openreview.net/forum?id=B1Yy1BxCZ}
>
> * Minor suggestions.
>
> We will update the plots, for example making it clear in Figure 5 that the y-axis are epochs, and will improve the text in the captions.

---

### Author Response · Authors · 2023-11-21
**Thanks to the reviewers for the comments, and how we addressed them**

We want to start by thanking the reviewers for their thorough revisions and insightful suggestions on how to improve our paper. We have ran two extra experiments that we think will help shed some light on some aspects of the SplitSGD procedure.

The first is a comparison between SplitSGD, SGD and Adam on ResNet on Cifar100. Our results show that, using the same parameters that we used for all other DNN experiments (q=0.25 and w=4) and the same set of starting learning rates, SplitSGD shows better performances than both methods for various choices of the initial learning rate (ranging from $0.1$ to $0.005$).

The second experiment is another sensitivity analysis, this time using ResNet on Cifar10. Similarly to what we observed in Figure 8, also in this setting we observe that substantial deviations from the default values for q and w ($0.25$ and $4$ respectively) do not impact the performance of the model significantly.

We hope these new results, together with the ones already in the paper, will convince the reviewers that the excellent performances of SplitSGD in many different settings is not due to a careful tuning of the hyperparamenters that we introduced, but are instead a consequence of the goodness of the Splitting Diagnostic in detecting stationarity.

---

### Decision · Action_Editor_HxpV · 2024-01-29

**Recommendation:** Accept with minor revision

**Comment:**

The paper has been reviewed by three reviewers: two recommend "leaning accept" (5Pef and ocBo), and one recommends "leaning reject" (dxDG). The main criticism of dxDG was that the simulations were not extensive enough. The authors have provided results on Cifar100, during the rebuttal phase, which i think is reasonable. However, these results should now be included in the final version of the paper.
As reviewers 5Pef and ocBo, I believe that the paper proposes an interesting idea. Although the available convergence results are not as strong as I would like (they rely on very strong assumptions), they are still of interest.

I thus recommend acceptance of the paper subject to the authors including (and discussing) the simulation results  presented in the rebuttal in the final version of the paper.

**Audience:**

SGD being used everywhere in machine learning, this paper should find a large audience.

**Claims And Evidence:**

The claims made in the paper are supported by some interesting theoretical results and extensive experiments.

---

> ### Author Response · Authors · 2024-02-16
> **Thanks to the Action Editor and Reviewers**
>
> We thank the action editor for their comments and suggestions and the recommendation to accept our paper. We are glad the action editor found our paper to be of interest to the community. We would also like to thank all the reviewers for their detailed feedback during the review process, which has helped us significantly improve our paper. We have addresses the comments made by the action editor and reviewers in the camera-ready submission.